

# Deforestation effects on soil erosion rates and soil physicochemical properties in Iran: a case study of using fallout radionuclides in a Chernobyl contaminated area

Maral Khodadadi[1,2]*, Christine Alewell[3], Mohammad Mirzaei[4], Ehssan Ehssan-Malahat[5], Farrokh

Asadzadeh[5], Peter Strauss[6], Katrin Meusburger[7]

[1]Nuclear Agriculture Research School, Nuclear Science and Technology Research Institute (NSTRI), 31485/1498, Iran
[2]Department of Geography and Regional Research, University of Vienna, Universitatsstraße 7, A-1010, Vienna, Austria
[3]Environmental Geosciences, University of Basel, Bernoullistrasse 30, CH-4056 Basel, Switzerland

[4]Nuclear Science and Technology Research Institute (NSTRI), Karaj, 31485/1498, Iran
[5]Department of Soil Science, Urmia University, Urmia, 5756151818, Iran
[6]Land is Water, Institute for Land and Water Management Research, Federal Agency for Water Management, Pollnbergstrasse 1, A-3252 Petzenkirchen, Austria
[7]Swiss   Federal   Institute   for   Forest,   Snow   and   Landscape   Research   (WSL),   8903   Birmensdorf,

Switzerland

*Correspondence to*: Maral Khodadadi (maral.khodadadi@univie.ac.at)

**Abstract.** Deforestation for farming and grazing purposes has become a global challenge. To study the impact of deforestation on soil erosion rates and soil physicochemical properties, Zarivar Lake watershed, Kurdestan Province, Iran, was selected. Converting the steep hillslopes naturally under oak forest to rainfed vineyards has been one of the most

common land-use changes in the area. We used $^{137}$Cs and $^{210}$Pb$_{ex}$ radionuclides and quantified the Chernobyl-derived $^{137}$Cs fallout with $^{239+240}$Pu. The soil samples were collected from two adjacent and similar hillslopes, one of which is under natural forest, while the other is under rainfed vineyard. Using $^{137}$Cs/$^{239+240}$Pu rates and a simple unmixing of the $^{137}$Cs sources indicated that 50.2±10.0% of $^{137}$Cs was Chernobyl-derived. The mean reference inventory values of $^{137}$Cs, $^{210}$Pb$_{ex}$, and $^{239+240}$Pu were estimated to be at 6152±1266, 6079±1511, and 135±31 Bq m$^{-2}$, respectively. At the forested hillslope, net soil

erosion rates based on $^{137}$Cs, and $^{210}$Pb$_{ex}$, techniques were estimated to be at 5.0 and 5.9 Mg ha$^{-1}$ yr$^{-1}$, respectively, resulting in Sediment Delivery Ratios (SDRs) of 96 and 70%. However, at the vineyard hillslope, the net soil redistribution rates were at 25.9 and 32.5 Mg ha$^{-1}$ yr$^{-1}$ for $^{137}$Cs and $^{210}$Pb$_{ex}$, respectively, resulting in respective SDRs of around 95 and 92%. Both $^{137}$Cs and $^{210}$Pb$_{ex}$ indicated that as a result of deforestation, soil erosion has increased by approximately five times. Percolation Stabilities (PS) in forest and vineyard topsoil are about 309 and 160 gr H$_2$O 600s$^{-1}$ classified as rapid and

moderate PSs, respectively. Rapid PS in forest soil implies high aggregate stability, whereas moderate PS in vineyard soils indicates that they are generally weakly-structured. All in all, the results of the present study revealed that deforestation and converting natural vegetation to cropland prompted soil loss and deteriorated physicochemical properties of the soil.

**KEY WORDS**: $^{137}$Cs; $^{210}$Pb$_{ex}$; $^{239+240}$Pu; Percolation stability, Soil erodibility; Land-use change.

# 1 Introduction

A rapidly increasing population, along with the tendency for a better standard of living, has induced the

need for intensive agricultural production in Iran. In order to provide food security for the population,





deforestation and converting forests to cultivated lands have increased drastically. In fact, over the past

50 years, the land-use changes in Iran have been more rapid than ever before and are expected to accelerate in the future (Emadodin et al., 2012). The area of natural forests in Iran has decreased from 19 million ha in the 1950s to 12.4 million ha in the 1990s (DEI, 2003). During the past 50 years, the area of Iran's cultivated land has expanded by more than five times, increasing from 2.6 million ha (Wilber, 1948) to 18.5 million ha (DEI, 2003). Simultaneously, more than 2.5 million ha of Iran's

agricultural land has converted to urban areas during the last decade, threatening food security. Consequently, deforestation and cultivation of marginal lands, including fragile upland ecosystems, have become more prevalent in the country. There is general agreement on the fact that this change in land-use on steep slopes leads to an increase in soil erosion (Collins et al., 2001; Nunes et al., 2011; Alatorre et al., 2012; Rudiarto and Doppler, 2013; Zhang et al., 2017; Nabiollahi et al., 2018) and a

decrease in soil quality and health (Nabiollahi et al., 2018). Many studies have shown that cultivation results in loss of organic matter (OM) (Vagen et al., 2006; Ozgoz et al., 2013; Karamesouti et al., 2015; Schweizer et al., 2017) which in turn influences the chemical, physical, and biological properties of the soil (Batlle-Bayer et al., 2010; Vinhal-Freitas et al., 2017; Zhang et al., 2017). In general, considerable impacts of deforestation on soils are biological degradation, soil compaction and physical degradation,

and accelerated erosion (Lal, 1989). Additionally, land-use change is considered to be one of the main causes of carbon emissions after fossil fuel consumption (Watson et al., 2000).

To study the impact of deforestation on soil physicochemical properties and soil erosion rates on steep slopes, Zarivar Lake watershed, Kurdestan Province, northern-west of Iran, was selected. The watershed is located in the Zagros forests in the Zagros mountain ranges, northern-west and west of the country.

Due to the dominance of oak species, these forests are known as western oak forests. Converting the steep hillslopes naturally under oak forest to rainfed vineyards has been one of the most common land-use changes in the area. The converted farmland is prone to erosion due to steep topography and the inappropriate land management practices (e.g., leaving the soil surface bare, and intensive tillage) in the past and the present. So far, only limited studies sought to quantify soil loss and changes in soil

properties in the Zarivar watershed (e.g. Ebrahimi-Mohammadi et al., 2012).

Fallout radionuclides (FRNs) mainly Caesium-137 ($^{137}$Cs), unsupported Lead-210 ($^{210}$Pb$_{ex}$) and in recent years also $^{239+240}$Pu, proved helpful in quantifying the impact of the deforestation on soil erosion



(Meusburger et al., 2016). $^{137}$Cs (half-life 30.2 years), is an anthropogenic isotope, produced by atmospheric testing of thermonuclear weapons during the 1950s and 1960s (i.e. global fallout) or the nuclear incidents like the one which took place in Chernobyl NPP in April-May 1986. On the contrary, $^{210}$Pb (half-life of 22.3 years) is a geogenic radionuclide, originated from the decay of gaseous $^{222}$Rn, a daughter of $^{226}$Ra (Mabit et al., 2008). A proportion of $^{222}$Rn produced from the lithogenic sources escapes and produces $^{210}$Pb which is deposited on the soil surface and is called "excess" or "unsupported" $^{210}$Pb ($^{210}$Pb$_{ex}$), because it is not in equilibrium with its parent $^{226}$Ra, therefore, $^{210}$Pb$_{ex}$ can be used as a soil redistribution tracer.

Generally, the global pattern of global-derived $^{137}$Cs fallout indicates that inputs are distributed between 160 and 3200 Bqm$^{-2}$ depending on latitude (UNSCEAR, 1969; Garcia Agudo, 1998). To estimate soil redistribution rates using $^{137}$Cs measurements, additional $^{137}$Cs fallout inputs which were originated from the Chernobyl incident must be taken into account. If $^{137}$Cs inventories are to be converted into annual soil erosion rates in the contaminated areas, the proportion of $^{137}$Cs Chernobyl fallout should be determined (Meusburger et al., 2018). To quantify the relative proportion of Chernobyl fallout to the total $^{137}$Cs inventory, plutonium (Pu) radioisotopes have recently been suggested (Alewell et al., 2017) and applied successfully (Alewell et al., 2014; Meusburger et al., 2016; 2018; 2020).

The two major anthropogenic radioisotopes of Pu isotopes, i.e. $^{239}$Pu [half-life of 24 110 - years] and $^{240}$Pu [half-life of 6561 years], are alpha-emitting actinides coming from nuclear weapon tests, nuclear weapon manufacturing, nuclear fuel reprocessing and nuclear power plant accidents (Ketterer and Szechenyi, 2008; Alewell et al., 2017). Globally, it was observed that the Pu 1950s-60s fallout is very similar to that of $^{137}$Cs distribution. However, in contrast to $^{137}$Cs, Pu is found in the non-volatile fraction of nuclear fuel debris released from reactor accidents like the one in Chernobyl accident. Hence, specific regions of Russia, Ukraine, Belarus, Poland, the Baltic countries, and Scandinavia were the geographic zones over which Pu Chernobyl fallout was deposited (Mietelski and Was, 1995). Therefore, Chernobyl Pu is unlikely to be found in other areas; as a result, it cannot be seen as a significant contributor to the total Pu activity deposited in Iran. Therefore, the relative proportion of Pu isotopes can reveal whether the area received additional $^{137}$Cs deposition from the Chernobyl accident or not. Pu radioisotopes have been applied successfully in Germany (Schimmack et al., 2001; 2002),





Switzerland (Alewell et al., 2014; Meusburger et al., 2016; 2018), Australia (Everett et al., 2008; Tims et al., 2010; Hoo et al., 2011; Smith et al., 2012; Lal et al., 2013), and China (Xu et al., 2013).

With this study, we aim to assess the impact of deforestation on physical degradation of soils in a case study in Iran. Conversion to farmland alters soil properties, and these changes may further enhance soil
erosion. An important factor in soil resistance to erosion is aggregate stability. Percolation Stability (PS) is a suitable method to assess aggregate stability (Mbagwu and Auerswald, 1999), with the advantage of less effort compared to the wet sieving method. Through rapid wetting, PS calculates the resistance of aggregates against slaking, thus triggering numerous subprocesses of erosion (Auerswald, 1995). Also, crusts contribute to run off and soil erosion (Lal, 1989; Valentin and Bresson, 1997; Fageria et al.,
2010). The crust formation is associated with low organic matter (OM) content and a high percentage of silt (Lal, 1989; Valentin and Bresson, 1997). As monitoring changes in permeability and soil aggregate stability are often time-consuming and costly; various indices were developed for predicting soil crusting using more available data such as soil texture and organic matter content (Udom and Kamalu, 2016). We compared two indices, namely, Sealing Index (SI) (Valentin and Bresson, 1997) and
Crusting Index (CI) (FAO, 1978).

The present study was undertaken to quantify the impact of deforestation on and soil loss and soil physicochemical properties in a representative region in Northwest Iran. The objectives of this study were (i) to estimate the proportion of $^{137}$Cs Chernobyl input using $^{239+240}$Pu in the study site, (ii) to quantify soil redistribution rates using $^{137}$Cs and $^{210}$Pb$_{ex}$ in two adjacent hillslopes under different land-
uses, and (iii) to evaluate the effects of deforestation on soil loss and soil physicochemical properties.

## 2 MATERIAL AND METHODS

### 2.1 Study area

Zarivar Lake (35° 33' 15" N and 46° 7' 25" E) is a shallow freshwater reservoir located approximately 15 km far from Iran and Iraq borders and 3 km northwest of Marivan city, Kurdestan Province, Iran (Fig.
1). The watershed was divided into 12 sub-watersheds (Fig. 2). "Z3" sub-watershed was selected for this study. The area of the sub-watershed is about 2.97 Km$^2$ with an average altitude of 1518 m a.s.l. and the landscape topography ranges from gentle to very steep slopes. The average annual precipitation and potential evaporation (using evaporation pan) in the area are 991.2 and 1626 mm, respectively (Iran



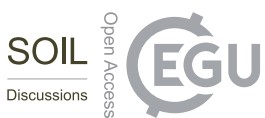

Meteorological Organization). The average annual, minimum and maximum air temperatures are 12.8, -
4.6 and 35.4°C, respectively (IMA, 2008).

Zarivar Lake is closed by mountains covered by deciduous oak forests of various species such as *Quercus spp.* (oaks), *Pistacia mutica* (wild pistachio), *Crategus spp.* and *Pyrus spp.* (Ghazanfari et al., 2004; Haidari et al., 2012). The forest is an open forest type. The ground flora normally presents a high degree of cover (up to 100%) which is exploited for grazing. According to the study of pollen in the lake cores, forests began to grow in the region around 5000 years ago (Van Zeist and Bottema, 1977).

The parent materials in ridges are metamorphic rocks of complexly folded black shale and minor metamorphosed limestone and sandstones (Geological survey and mineral exploration of Iran). However, parent materials in the lowlands consist of Quaternary sediments, namely, lacustrine deposits which include sand, marl, and loam (Geological survey and mineral exploration of Iran). According to soil taxonomy, the soils of the watershed are Typic Haploxeralfs and Typic Haploxerepts (Soil Taxonomy, 2014) and based on WRB, they are categorised as Haplic Luvisols and Haplic Cambisols orders (WRB, 2014). The dominant top soil textures are silty loam, and loam and the stone content differs profoundly (IMA, 2008).

The most dominant irrigated crops in the sub-watershed are alfalfa, tobacco, strawberry, vegetables, etc. as well as scattered orchards of walnut. Also, rainfed crops such as vineyard and cereal crops, namely, wheat, barley, and lentil, are cultivated in the area. The vineyards are extensively cultivated on sloping lands. Generally, different soil erosion types, including sheet, rill, stream bank, gully are found in the watershed. However, conservation measures like terraces and check dams have been constructed in some parts of the watershed during the last decade.

## 2.2 Soil redistribution rates using FRNs

### 2.2.1 Soil sampling design

FRNs as soil erosion tracers are normally used by comparing their inventories in sites affected by soil redistribution processes with their baseline inventory at a reference site (Mabit et al., 2008, 2012). A crucial step that should be taken when using FRN techniques is to find an undisturbed reference site (Mabit *et al.* 2008, 2012). Here an undisturbed, flat site, having a cover of perennial grass, in clearance between trees, was selected. This reference site with an elevation of 1397 m is located ca. 800m far

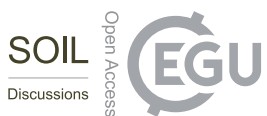

from the study sites (Fig. 2). Eleven bulk samples were collected to determine the initial inventory of FRNs at the reference site. For collecting samples, manual soil column cylinder augers (inner diameter of 9 cm) were designed and built for stony soils. Its cutting edge was made up of titanium so that it

could be able to penetrate the stony soils easily. A sectioned core was collected at 2 cm increments using a scraper plate sampling device designed by Campbell et al. (1988).

At the two studied hillslopes, a multi-transect sampling approach was adopted. In each site, a total of 33 bulk soil cores have been gathered along the three parallel transects in the main slope direction (Fig. 1). Soil sampling was performed up to 30 and 40 cm depth in forest and vineyard hillslopes, respectively to

ascertain that all FRNs in the soil profile were included. A sectioned core at 5 cm increments was also collected in an erosional site in each hillslope.

### 2.2.2 Soil sample pre-treatment and Radioisotopic analysis

Soil samples were air-dried, disaggregated, sieved to <2 mm, grounded and then homogenized. Soil samples were analyzed for $^{137}$Cs and $^{210}$Pb and $^{226}$Ra by gamma spectrometry using N-type HPGe

detector. Gamma spectrometry measurements were performed in the Iranian Nuclear Medicine Research Institute.

Before analyzing $^{210}$Pb, sub-soil samples were sealed for one month to achieve equilibrium between $^{226}$Ra and its daughter $^{214}$Pb. $^{137}$Cs, total $^{210}$Pb and $^{214}$Pb activities were determined from the net peak areas of gamma rays at 661.6, 46.5 and 352 keV, respectively. The counting times ranged from 12 to 24

h with a precision of 5% to a maximum of 20% at the 95% level of confidence.

To determine the proportion of Cs Chernobyl fallout in the samples, 39 bulk and increment samples were selected, which included reference bulk samples, reference increment samples, a transect in forest and increment samples of an erosional site. According to the method described by Ketterer et al. (2012), the samples were prepared for the analysis of Plutonium isotopes. The measurement of Plutonium

isotopes ($^{239+240}$Pu) was conducted using a Thermo X Series II quadrupole ICP-MS instrument at the University of Cadiz, Spain. The ICP-MS instrument was equipped with a high-efficiency dissolving sample introduction system. A detection limit of 0.1 Bq kg$^{-1}$ for $^{239+240}$Pu was obtained for samples of nominal 1 g of dry-ashed material; for $^{239+240}$Pu activities> 1 Bq kg-$^1$. The measurement error was 1– 3%. $^{239}$Pu and $^{240}$Pu masses existing in the sample (which were determined by isotope dilution



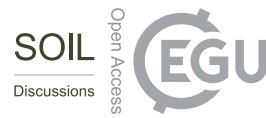

calculation) were presented as the summed $^{239+240}$Pu activity so that it could correspond with alpha spectrometric determinations of Pu activity. The $^{240}$Pu/$^{239}$Pu atom ratios were determined through the same analytical run. Data quality was evaluated through the analysis of preparation blanks (soils or rocks devoid of Pu), duplicates, and control samples with known $^{239+240}$Pu activities.

The soil redistribution rates were, therefore, estimated using the Diffusion and Migration Model
(DMM) (Walling et al., 2002, 2014) in forested hillslope, Mass Balance Model II (MBM II) (Walling et al., 2002, 2014) in cultivated hillslope, and Modelling Deposition and Erosion rates with RadioNuclides (MODERN; Arata et al., 2016a, 2016b).

**2.3 Soil sampling and soil physicochemical properties**

For this study soil samples were collected from two adjacent similar hillslopes, one of which is under
natural forest while the other hillslope is under rainfed vineyard, which was deforested in 1981. The hillslopes were selected having similar slope gradient, length, and exposures (Fig. 2). Disturbed and undisturbed soil samples (for the latter we used a 5 cm diameter and 5 cm high steal ring) were collected along transects at six and seven points in the vineyard and forested hillslope, respectively. Both undisturbed and disturbed samples were gathered from two depths of 0-20 ad 20-40 cm. The
disturbed samples were air-dried, disaggregated and sieved to <2 mm. Soil pH (McLean, 1982) and Electric Conductivity (ECe) (Rhoades, 1982) were determined in the saturated paste and saturated paste extract, respectively. Soil texture, or Particle Size Distribution (PSD) (wet sieving and pipette method, Gee and Bauder, 1986), and bulk density (BD) (core method; Blake and Hartge, 1986) were measured. Porosity (*f*) of the soils was calculated using soil bulk density (Danielson and Suterland, 1986). The soil
moisture at field capacity (FC) and the wilting point was (PWP) determined by a pressure plate apparatus (Klute, 1986) at 33 and 1500 kPa, respectively. The Available Water Capacity (AWC) was calculated by subtracting FC and PWP. Percolation stability was measured in a laboratory test with air-dried 1 to 2 mm aggregates (Siegrist et al., 1998) at the Federal Agency for Water Management Austria, Vienna, Austria. In this method, the water is percolated through a small plexiglass tube, filled with
sample, with a constant hydrostatic head. The amount of percolated water during 600s was used to calculate PS values.



Total Organic Carbon (TOC) (Walkley-Black method; Nelson and Sommers, 1982), Carbonate Calcium Equivalent (CCE) (Jackson, 1958) and Cation Exchange Capacity (CEC) (determined using Rhoades, 1982 method) were also measured using standard methods. TN was determined by Kjeldahl method

(Bremner, 1996). The soil C:N ratio was calculated by dividing the SOC concentration by the TN concentration. The soil organic carbon stock (SOCs) was computed with the equation proposed by Hiederer and Köchy (2011) as well. Soil permeability was measured using double-ring infiltrometers, and a soil profile was also studied in each hillslope (Scholten, 1997).

Additionally, soil erodibility (K) factor was estimated using Wischmeier and Smith (1978). The

equation takes into account the OM content, aggregate stability, infiltration rate, and particle size distribution. Crusting hinders germination of feeble seedlings and reduces infiltration, which on steep slopes, can lead to runoff and soil erosion. To assess soil's susceptibility to crusting, soil physical quality indices were also computed, which included Sealing Index (SI) (Valentin and Bresson, 1997) and Crusting Index (CI) (FAO, 1978) using equations 1, and 2, respectively.

$$SI = \frac{OM \times 100}{Silt + Clay} \tag{1}$$

Value of the sealing index, if less than 5%, is considered high crusting or sealing risk, whereas a value more than 9% indicates a low sealing risk and 7% is regarded as the threshold value (Udom and Kamalu, 2016).

$$CI = \frac{1.5(\%fine\ Silt + 0.75 coarse\ Silt)}{\%Clay + 10(\%OM)} \tag{2}$$

The values over 1.6 for CI represent that soils are prone to intense crusting, whereas values below 1.2

are implying low crusting risk.

Basically, we performed a statistical comparison between topsoils and subsoils properties of the two land-use types. This was done by one-way analysis of variance (ANOVA). All statistical analyses were performed using SPSS software ver. 22 (IBM, Armonk, NY, USA).





## 3 Results

### 3.1 Assessment of soil redistribution rates estimated by FRNs

### 3.1.1 FRN baseline inventories and $^{137}$Cs Chernobyl fallout proportion in the study area

The depth profiles at the reference site of three FRNs showed an exponential decrease with depth (Fig. 3). As illustrated in Fig. 3 b, c, d, the upper 12 cm of the soils contained 91.5%, 91.7%, and 90.4% of the $^{137}$Cs, $^{239+240}$Pu, and $^{210}$Pb$_{ex}$ inventory, respectively. The coefficient of variations (CV) for all three

FRNs were around 20% at the reference site (21, 23, 25% for $^{137}$Cs, $^{239+240}$Pu and $^{210}$Pb$_{ex}$, respectively) (Fig. 4). The mean reference inventory values of $^{137}$Cs, $^{210}$Pb$_{ex,}$ $^{239+240}$Pu were estimated to be 6152±1266, 6079±151, and 135 ±31 Bq m$^{-2,}$ respectively (Fig. 4).

The $^{137}$Cs value obtained in our study site was substantially higher than those reported for reference inventories in Iran so far (Table 1). It was hypothesized that the study site received Chernobyl-derived

$^{137}$Cs fallout. The CHELSA (Climatologies at high resolution for the earth's land surface areas) dataset (Karger et al., 2017) confirmed that the study site received around 150, and 100 mm rainfall in April and May 1986, respectively (the time when the radioactive Chernobyl cloud travelled across Iran; Fig. 5). Also, depth profiles of $^{137}$Cs in proximate (< 1 km distance) lake cores showed two distinct peaks (supplementary Fig. 1), which points out to the influence of $^{137}$Cs Chernobyl fallout.

At our study site, the average atomic ratio of $^{240}$Pu/$^{239}$Pu was 0.184 ± 0.020 ranging from 0.121 to 0.262 (Fig. 6). In addition, to determine the proportion of the $^{137}$Cs Chernobyl input at our study site, the mean activity ratio of $^{137}$Cs/$^{239+240}$Pu from global originated fallout was employed using the value reported by Hodge et al. (1996) at 38.4 (as of 1 July 1994) after values being decay-corrected. The Chernobyl $^{137}$Cs was derived from subtraction of reference site inventories from those calculated using the above-

mentioned ratio. The $^{137}$Cs Chernobyl input in the study site was estimated to be at 50.2±10.0 %.

### 3.1.2. Fallout radionuclides inventories and soil redistribution rates at slope transects

For $^{137}$Cs and $^{210}$Pb$_{ex,}$ $^{239+240}$Pu the MODERN and the DMM were applied at the forested site, while at the vineyard site, the MBM without tillage component was used (Table 2). The mean and standard deviation of $^{239+240}$Pu inventory was at 130.9±54.5 Bq m$^{-2}$ at the forested hillslope, $^{137}$Cs inventories

were at 5389.9±2227.9 and 4646.6±1921.8 Bq m$^{-2}$, respectively for forested and vineyard hillslopes and $^{210}$Pb$_{ex}$ inventories at 4068.3±2345.8 and 3990.1±2892.2 Bq m$^{-2}$, respectively for forested and vineyard





hillslopes (Fig. 4). The majority of sampling points on the steep forested slope and cultivated field showed a lower inventory compared to those of the reference site for each FRN, implying the dominance of erosive processes. At specific sampling points, $^{210}Pb_{ex}$ was below the detection limit.

Both $^{137}Cs$ and $^{210}Pb_{ex}$ showed moderately lower average inventories in vineyard hillslope.

At forested hillslope, net soil erosion rate estimated by MODERN for $^{210}Pb_{ex}$ (at 5.9 Mg ha$^{-1}$ yr$^{-1}$) was slightly more than that of $^{137}Cs$ (at 5.0 Mg ha$^{-1}$ yr$^{-1}$) and $^{239+240}Pu$ (at 1.7 Mg ha$^{-1}$ yr$^{-1}$), yielding respective SDRs of 96, 70 and 66% (Table 2). By contrast, the ones estimated by DMM were considerably different for these two FRNs, standing at 2 and 5.1 Mg ha$^{-1}$ yr$^{-1}$ for $^{137}Cs$ and $^{210}Pb_{ex}$,

respectively. At the vineyard hillslope, the net soil redistribution rates estimated by MBM II were at 25.9 and 32.5 Mg ha$^{-1}$ yr$^{-1}$ for $^{137}Cs$ and $^{210}Pb_{ex}$ respectively, which was about five times more than those estimated by MODERN at the forested hillslopes, resulting in respective SDRs of around 95 and 92% (Table 2). In general, SDRs calculated in both hillslopes using different models were profoundly high, indicating that most of the eroded soil has been removed from the hillslopes.

The correlation between inventories of $^{210}Pb_{ex}$ and $^{239+240}Pu$ with $^{137}Cs$ along a transect in forested hillslope was examined (Fig. 7). A stronger positive correlation was observed between $^{137}Cs$ and $^{239+240}Pu$ (ca. $r^2=0.82$) than between $^{137}Cs$ and $^{210}Pb_{ex}$ (ca. $r^2=0.56$). Also, the soil redistribution rates of three FRNs in a forested transect were estimated using the MODERN and DMM (Fig. 8). DMM estimated soil losses for $^{210}Pb_{ex}$ were higher than those of $^{137}Cs$ in most sampling points; yet MODERN

estimated soil losses and gains for $^{239+240}Pu$ were by far the highest along the transect (Fig. 8).

The sectioned profiles of $^{137}Cs$, $^{210}Pb_{ex}$ and $^{239+240}Pu$ in an erosional site in the second transect (mid-transect) of forested hillslope were compared (Fig. 9 a, b, and c). Total inventories of $^{137}Cs$, $^{210}Pb_{ex}$ and $^{239+240}Pu$ were 5095, 4670 and 19 Bq m$^{-2}$, respectively, indicating inventory changes of -17, -23 and -86%. While inventory changes of $^{137}Cs$ and $^{210}Pb_{ex}$ were comparatively similar, that of the $^{239+240}Pu$ was

notably high (Fig. 9 a, b, and c). Depth distribution of $^{137}Cs$ and $^{210}Pb_{ex}$ at an erosional vineyard site was investigated (Fig. 9 d and e). Total inventories of $^{137}Cs$ and $^{210}Pb_{ex}$ were 3807 and 2400 Bq m$^{-2}$, respectively, resulting in inventory changes of -38 and -61%.



## 3.2. Change of physicochemical properties of soils due to land-use change and soil erosion

Deforestation and land-use change from natural vegetation to agriculture have significantly increased
ECe, pH, BD, and the K factor, whereas they have significantly decreased OM, TN, C:N, CEC, AWC,
$f$, and PS in surface soil (Table 3). Likewise, there was a significant rise in pH, BD, and K in the
subsurface soil of vineyard compared to the subsoil of forest while clay, OM, TN, C:N, CEC, AWC, $f$,
and PS experienced a notable decline (Table 3).

OM and TN in both depths were reduced in the vineyard compared to the forest; however, this
reduction was greater in the surface layer (Table 3). According to Table 3, the average OM content of
the topsoil of the forest is twice as much as the dry farming vineyard i.e. 2.8 and 1.4%, respectively,
which correspond to a carbon stock of 28.8 and 19.8 Mg ha$^{-1}$, respectively. Similarly, the OM content in
forest and vineyard subsoil were 1.7 and 1.2%, respectively, indicating a carbon stock of 19.5 and 15.2
Mg ha$^{-1}$, respectively. The C:N ratio dropped with depth in both soils, most probably as a result of lower
OM turnover (Table 3). Moreover, bulk density of the vineyard topsoil was significantly greater than
that of forest (1.09 kg m$^{-3}$ and 1.37 kg m$^{-3}$ in the forest and vineyard topsoil, respectively) (Table 3).
Similarly, following the deforestation, the porosity of the surface and subsurface soils under vineyard
declined by around 11 and 10%, respectively.

Mbagwu and Auerswald (1999) classified PS into three classes, namely, rapid (>250 gr H$_2$O 600s$^{-1}$),
moderate (250-150 gr H$_2$O 600s$^{-1}$), and slow (<150 gr H$_2$O 600s$^{-1}$). PS in forest and vineyard topsoil
were about 309 and 160 gr H$_2$O 600s$^{-1}$ which classified as rapid and moderate PSs, respectively. Rapid
PS in forest soil implies high aggregate stability reflecting the higher OM content, whereas moderate PS
in vineyard soils indicates that they are generally weakly-structured. Furthermore, two indices were
calculated to evaluate the surface crusting risk. The sealing index (SI) in all soil samples was less than
5% indicating a high risk of sealing. Crusting Index (CI) in both topsoil and subsoil of the forest showed
a low sealing risk (<1.2), while it was considered to have a moderate sealing risk in surface and
subsurface soils of the vineyard (1.2-1.6). K-values in forest surface and subsurface soils were 0.044
and 0.054, respectively, whereas respective values for surface and subsurface vineyard soils were 0.051
and 0.067 (Table 3).





## 4. discussion

### 4.1 Assessment of soil redistribution rates estimated by FRNs

#### 4.1.1 FRN baseline inventories and $^{137}$Cs Chernobyl fallout proportion in the study area

The depth profiles at the reference site of three FRNs showed an exponential decrease with depth (Fig. 3). $^{137}$Cs is mostly associated with the fine mineral fraction (Lujaniene et al., 2002; Qiao et al., 2012) while $^{239+240}$Pu in soils exhibits strong binding to organic matter and sesquioxides. As such, both FRNs show a slight migration into the upper subsurface horizons. $^{210}$Pb$_{ex}$ is absorbed by both, organic matter and clay-sized mineral particles, and seem to show less migration into subsurface horizons. Evidently, $^{210}$Pb$_{ex}$ does have constant fallout while the other two FRNs don't (Mabit et al., 2008), which can be another reason for its less immigrations into subsurface horizons.

The mean reference inventory value of $^{137}$Cs was substantially higher than those reported for reference inventories in Iran so far (Table 1). However, it should be noted that our study site has a 2-3 times higher MAP as compared to the study sites investigated so far. FRN fallout is strongly related to MAP and is the main cause for high global$^{137}$Cs fallout (Meusburger et al., 2020). Owing to substantial amounts of rainfall after Chernobyl incident (CHELSA dataset; Karger et al., 2017), and the presence of two peaks in depth profiles of $^{137}$Cs in proximate lake cores (supplementary Fig. S1), a hypothesis was formed suggesting that the study site had been contaminated by Chernobyl-derived $^{137}$Cs fallout. Although other inventories of the $^{137}$Cs in different parts of Iran have not shown unusually high inventories (e.g. Afshar et al., 2010; Ayoubi et al., 2012; Rahimi et al., 2013; Khodadadi et al., 2018), moderately high inventories around 4000 Bq m$^{-2}$ observed in North of Iran (Gharibreza et al., 2020) were relatively compatible with our findings in the northwest of Iran (Table 1).

The $^{240}$Pu/$^{239}$Pu atom ratio of global fallout in mid-latitudes of Northern Hemisphere is $0.180 \pm 0.014$ (Kelley et al., 1999) while the ratio of Chernobyl fallout is between 0.37 to 0.41 (Muramatsu et al., 2000; Ketterer et al., 2004). Thus, our measured $^{240}$Pu/$^{239}$Pu ratios (with average of ca $0.184 \pm 0.020$) are in accordance with the integrated atmospheric fallout value of $0.18 \pm 0.014$ (Kelley et al., 1999), implying that the source of Pu is a result of the global atmospheric fallout (Fig. 6). The proportion of the $^{137}$Cs Chernobyl input at our study site was estimated to be at $50.2 \pm 10.0$ %. In comparison to the average value of Chernobyl contribution in the Swiss Alps which was found to be between 75%



(Meusburger et al., 2018) to 80% (Schaub et al., 2010; Alewell et al., 2014), the contribution found in our site was notably less.

The mean reference inventory value of $^{210}Pb_{ex}$ stood at 6079±1511 Bq m$^{-2}$ which was moderately higher than the inventory of $^{210}Pb_{ex}$ at 5825±297 Bq m$^{-2}$ in central Iran with mean annual precipitation of 330 mm (Khodadadi et al., 2018). The annual deposition flux of $^{210}Pb_{ex}$ was estimated to be 205 Bq m$^{-2}$ yr$^{-1}$ for the study area, the amount of which is within the range of the reported global annual deposition fluxes of $^{210}Pb_{ex,}$ i.e. from 23 to 367 Bq m$^{-2}$.

No data has yet been published on $^{239+240}Pu$ for Iran. The estimated reference inventory of 135 ±31 Bq m$^{-2}$ is considerably higher than the values reported for reference sites in Europe, China and Australia (Table 4). This deviation of $^{239+240}Pu$ inventory in our reference site from those values might be attributed to high initial bomb-derived deposition in the study site in 1953–1964, and the differences in the locations and the climates. However, measurements from other parts of the country are required to

confirm the hypothesis of a high initial bomb-derived fallout.

The coefficient of variations (CV) for all three FRNs were around 20% at the reference site (Fig. 4). Indeed, there is an inevitable spatial variability of FRNs inside any reference site; however, due to interception of fallout by the tree canopies and their trunks in the forest, a forested reference site, in particular, may represent relatively higher CVs (Wallbrink and Murray, 1996). Such high CVs have also

been reported in the literature; Nagle et al. (2000), for instance, reported a CV of 28% for forest and coffee reference sites in tropical mountains. Sutherland (1991) also found CVs ranging from 17.9 to 31.8 in reference sites of Saskatchewan, Canada. However, the $^{210}Pb_{ex}$ reference inventory values represented a higher spatial variability with a CV of 25%, which was in agreement with Meusburger et al. (2016) who reported a relatively higher CV for $^{210}Pb_{ex}$ (ca. 28%) than $^{137}Cs$ (ca. 18%) and $^{239+240}Pu$

(ca. 15%). Teramage et al. (2015) presented a CV of 28% for $^{210}Pb_{ex}$ at a coniferous forest in Japan.

As the key assumption of FRNs method is the homogeneity of the initial fallout in the reference site, the heterogeneity can obscure or even compromise the usage of this method (Haugen, 1992; Golosov et al., 2008; Mabit et al., 2013). As highlighted by Sutherland (1996) and Mabit et al. (2012), the allowable CV was estimated to be 20% for FRNs at 90% level of confidence, with which the selected reference

site here agreed, that is to say, our CV was approximately 20% as well. Moreover, 20% CV of $^{239+240}Pu$ confirmed that surplus $^{137}Cs$ Chernobyl fallout distributed uniformly in the study area. Unusually high



heterogeneity of $^{137}$Cs reference inventories was, however, found in the Swiss alpine areas (Alewell et al., 2014) which was essentially because of partial snow coverage at the end of April-May, 1986 and that most likely caused heterogeneous redistribution during the snowmelt process. However, unlike
Swiss Alps, the possibility of snow coverage in the so-called time period in the study area was unlikely.

#### 4.1.2. Fallout radionuclides inventories and soil redistribution rates at slope transects

The net soil erosion rates estimated by DMM at forested hillslope were at around 2 and 5.1 Mg ha$^{-1}$ yr$^{-1}$ for $^{137}$Cs and $^{210}$Pb$_{ex}$, respectively. Different values were reported for the net soil erosion rate at forested areas in the literature. For instance, by using $^{210}$Pb$_{ex,}$ Gaspar et al. (2013) estimated a mean soil
redistribution rate of 1.3 to 1.7 Mg ha$^{-1}$ yr$^{-1}$ at the slope gradient of 24% in the Mediterranean oak forest, Spain. Wakiyama et al. (2010) reported soil erosion magnitudes obtained by $^{210}$Pb$_{ex}$ ranging from 0.65 to 1.24 Mg ha$^{-1}$ yr$^{-1}$ in forested hillslopes with a slope gradient of around 40% in Japan. These values were nonetheless slightly lower than our estimated values.

At the vineyard hillslope, the net soil erosion rates estimated by MBM II were at 25.9 and 32.5 Mg ha$^{-1}$
yr$^{-1}$ for $^{137}$Cs and $^{210}$Pb$_{ex}$, respectively. Both FRNs estimated that as a result of deforestation, soil erosion has increased by approximately 5 times. Gharibreza et al. (2013) who used $^{137}$Cs method showed that following deforestation in a watershed in Malaysia, soil loss magnitude surged by 7-13 times, which was compatible with our findings.

It should be noted that different FRNs inherently account for diverse time spans (Meusburger et al.,
2016). Relatively high $^{137}$Cs Chernobyl input at the site under investigation (~50%) reveals that $^{137}$Cs is more likely to be an indicator of a medium-term soil erosion process, i.e. mainly from 1986 onwards. However, the time frame captured by $^{239+240}$Pu has been recorded from mid-1960s onward. Moreover, although $^{210}$Pb$_{ex}$ inventories are influenced by the last 100 years of fallout deposition, due to its ongoing fallout through time, it is more sensitive to the two last decades' erosive events causing soil
redistribution (Porto et al., 2013, 2014). At forested hillslope, the net soil erosion rate estimated by $^{239+240}$Pu was far less than that of $^{137}$Cs (Table 2) suggesting that a higher soil loss might have occurred after 1986 compared to 1963 to 1986 time-window. Similarly, of $^{137}$Cs was just slightly less than that of $^{210}$Pb$_{ex,}$ which could be a reflection of last decades' climate variabilities.





A trend analysis of average annual precipitation of two adjacent synoptic stations, i.e. Sanandaj and
Marivan was done using the Mann-Kendall test and Sen's slope estimates (Salmi, 2002). Mean annual
precipitation of both stations has shown a clear downward trend from 1959 onwards. Additionally,
Balling et al. (2016) showed an increasing trend for extreme precipitation events over the period 1951–
2007. Khodadadi et al. (2018) also reported a higher amount of soil loss derived by $^{10}Pb_{ex}$ compared to
that of $^{137}Cs$. As pointed out by Porto et al. (2013, 2014), $^{210}Pb_{ex}$ inventories are more likely to reflect
soil redistribution caused by erosive events over the last 15 to 20 years. As found in similar studies,
however, measurement uncertainties of $^{210}Pb_{ex}$ may affect the estimated soil loss magnitudes to some
extent.

The soil redistribution rates of three FRNs in a forested transect were estimated using both the
MODERN model and DDM (Fig. 8). Results obtained for $^{137}Cs$ by the MODERN represented a wider
range of soil redistribution magnitude, while the DMM outputs tended to level off the extreme soil
redistribution rates, which were in accordance with Meusburger et al. (2018).

Inventory changes in sectioned profiles of $^{137}Cs$, $^{210}Pb_{ex}$ and $^{239+240}Pu$ in an erosional site at forested
hillslope were relatively different (Fig. 9 a, b, and c). In other words, while inventory changes of $^{137}Cs$
and $^{210}Pb_{ex}$ (-17, and -23 respectively) were quite identical, the inventory change of the $^{239+240}Pu$ (-
86%), was notably high (Fig. 9 a, b, and c). Supposedly, changes in microtopography of the hillslope
over time might have brought about such discrepancies in FRNs inventories. Depth distribution of $^{137}Cs$
and $^{210}Pb_{ex}$ at an erosional vineyard site confirmed that an erosion process has occurred on the site (Fig.
9 d and e). Yet, $^{210}Pb_{ex}$ showed a higher rate of soil erosion. Such an incongruity was also found by
Wakiyama et al. (2010) when applying $^{137}Cs$ and $^{210}Pb_{ex}$ in Japan. This can suggest that erosion has
increased at the sampling point in response to the climate variabilities over the last decades. On the
whole, changes in land-use and land management (Gaspar et al. 2013; Zhang et al. 2006),
microtopography and the rainfall characteristics during the different timescales might have a significant
effect on the estimated soil erosion rates using different FRNs.

### 4.2. Change of physicochemical properties of soils due to land-use change and soil erosion

Soil organic matter content (OM), a key quality index of soil (Pathak et al., 2004), influences the
chemical, physical, and biological properties of soil, therefore directly impacting microbial and plant





growth. OM and TN in both depths were reduced in the vineyard compared to the forest (Table 3). Puget and Lal, 2005; Khormali et al., 2009; Hernanz et al., 2009; and Nabiollahi et al., 2018 also reported a considerable reduction in SOM after the land-use change, which were congruent with our

findings. This reduction can partly be due to a lower litter turnover and biomass exiting from the hillslope. In addition, tillage operations which are done to eliminate weed growth and conserve moisture are probably going to increase organic-matter oxidation (i.e. mineralisation) by breaking up the soil aggregates. In addition, factors such as low vegetation cover, less shading, increased soil temperature, tillage practices, and subsequent susceptibility of soils to erosion after deforestation are key factors

prompting the loss of OM (Carter et al., 1998; Gregorich et al., 1998; Six and et al., 2000). Also, the breaking down of the larger aggregates to smaller ones following deforestation increases the loss of OM (Nardi et al., 1996). Forest soils had higher C:N ratio in both depths. This trend was comparable to those reported by Puget and Lal (2005) and Franzluebbers et al. (2000). In fact, the ratio mostly corresponds to the amount of plant residues entering the soil organic matter pool (Diekow et al., 2005; Puget and

Lal, 2005). The C:N ratio dropped with depth in both soils, most probably as a result of lower OM turnover (Table 3).

Bulk density of the vineyard topsoil was significantly greater than that of forest (Table 3). In fact, as stated above, deforestation leads to the reduction of OM; consequently, aggregate stability is reduced, which in turn gives rise to soil aggregate rupture, resulted from slaking. In addition, soil compaction,

which is the result of tillage operation, does increase subsequently (Celik, 2005; Puget and Lal, 2005). This will lead to a reduction of $f$ and increase of the BD (Celik, 2005; Bahrami et al., 2010). Similarly, available water content (AWC) of forest topsoil was higher by approximately 5%. The vineyard subsoil retained as much AWC as forest subsoil.

PS in forest and vineyard topsoil were classified as rapid and moderate PSs, respectively. Rapid PS in

forest soil implies high aggregate stability reflecting the higher OM content, whereas moderate PS in vineyard soils indicates that they are generally weakly-structured. It is also believed that fungal hyphae play an important role in aggregate stability under forest (Ternan et al., 1996). Mbagwu and Auerswald (1999) also reported rapid PS in forest soils, and slow to moderate PS values in bare fallows and cultivated plots. They concluded that PS positively correlated with OM. Notable is the PS of topsoil

samples in both land-uses were higher than those of their subsoils. The PS of topsoil sample of the



vineyard was roughly as much as that of the subsoil sample forest, implying that it was affected by the land-use change history. From among all properties shown in table 3, the highest coefficient of variability occurred in PS, which was notably higher in vineyard surface and subsurface soils.

Microaggregates resulting from aggregate breakdown clog large pores, thus decreasing infiltration rate and increasing runoff generation (Auerswald, 1995). The sealing index (SI) in all soil samples was less than 5% indicating a high risk of sealing which can be due to a high percentage of silt (Valentin and Bresson, 1997; Lal, 1989). However, SI was significantly higher in both topsoil and subsoil of forest owing to higher OM content. Udom and Kamalu (2016) reported that SI varied from 2.40 to 3.10 in topsoil and from 2.17 to 4.76 in the subsoil of tropical soil susceptible to seasonal flooding, meaning that the risk of sealing was considerably high in their investigated sites. In contrast, Crusting Index (CI) in both topsoil and subsoil of the forest showed a low sealing risk, while it was considered to have a moderate sealing risk in surface and subsurface soils of the vineyard. In other words, although SI indicated a high risk of sealing for all studied soils, CI the suggested that risk of crusting in both surface and subsurface soils of the vineyard were more than those of forest soils. This implies that the CI can be an advantageous index over SI as it can not only take the particle size classes of silt (i.e. fine and course silt) into consideration but also better differentiate between the soils with different OM.

The K factor depends on particle size distribution, permeability, organic matter content, and structure. K-values in forest surface and subsurface soils were lower than those for vineyard soils. The reported K factor in China ranged from 0.007 to 0.043 (Wang et al., 2013), which is similar to the K-values in the USA varying from 0.004 to 0.063 (Wischmeier and Smith, 1978; Romkens et al., 1997). Comparing these values with our site, we can conclude that there are high values of the soil erodibility in the study area. In other words, the soils of both land-uses are extremely prone to soil erosion owing to the high amount of silt. Following the land-use change, K factor value in the vineyard has increased up to 14% in topsoil and around 20% in the subsoil, which was in agreement with Boix-Fayos (2001); Khormali et al., (2009) and Evrendilek et al. (2004). Based on the study of Sanchez-Maranon et al. (2002), following the land-use change, the K-value has risen up to around 59% in southern Spain. Also, Celik (2005) studied the impact of cultivated land on soil properties in the south of Turkey and reported that cultivation practices have increased the amount of K-value by 2.4.



Even though significant differences in clay content in forested and deforested topsoil sites were not
observed, the clay content in forest topsoil was slightly less than that of cropland. On the other hand, the
clay content of forest subsoil was considerably higher, reflecting that clay lessivation might have
occurred in the forest soils. Besides, subsoils characterized by higher clay content may be drawn to the
surface by regular deep ploughing, resulting in the minor increase of clay content in the cropland
topsoil. All in all, the results of the present study revealed that deforestation and converting natural
vegetation to cropland prompted soil degradation, deteriorating physicochemical properties of the soil.

**5. Conclusion**

Forests and forest soils store carbon, and therefore play a decisive role in mitigating climate change
impacts. In Iran, over the past decades, a rapidly growing population has induced an increasing demand
for food, so one of the most rapid land-use change, i.e. conversion of land under natural vegetation into
arable lands, has been occurring. The present study was undertaken to quantify the impact of
deforestation on soil loss with fallout radionuclides and soil physicochemical properties.

The deposition of the $^{137}Cs$ fallout from different sources (Chernobyl fallout vs. global fallout) can be
determined via $^{239+240}Pu$ isotopes if Pu origins exclusively from global fallout. The $^{240}Pu/^{239}Pu$ atom
ratios in the reference samples confirmed the global fallout origin of Pu. From the $^{137}Cs/^{239+240}Pu$ ratio,
it was evident that half of the $^{137}Cs$ found in the site was Chernobyl-derived. The mean reference
inventory of $^{137}Cs$ at $6152\pm1266$ Bq m$^{-2}$ was relatively higher than previous reports in different parts of
Iran, which might be due to the location of the site in Iran (the site was closer to the Chernobyl site than
other measured sites in Iran), with higher mean annual precipitation, and relatively high rainfall after
Chernobyl incident. The mean reference inventories of $^{210}Pb_{ex}$ estimated at $6079\pm1511$ Bq m$^{-2}$ was in
accord with the reported value in the central part of Iran. The mean reference inventory of $^{239+240}Pu$ at
$135\pm31$ Bq m$^{-2}$ was higher than values reported in other parts of the globe, which might be attributed to
high initial bomb-derived deposition in the study site during 1953–1964, and the location and climate
diversity. Nevertheless, measurements from other parts of the country are required to confirm the
hypothesis of a high initial bomb-derived fallout.

Both $^{137}Cs$ and $^{210}Pb_{ex}$ radionuclides indicated that deforestation has increased annual soil loss by about
five times. At the forested site, net soil losses for $^{137}Cs$ and $^{210}Pb_{ex}$ were around 5.0 in and 5.9 Mg ha$^{-1}$





yr$^{-1}$ (using MODERN model), respectively, and at the vineyard hillslope, they were about 25.9 and 32.5 Mg ha$^{-1}$ yr$^{-1}$ (using DMM), respectively. Notable is that the values obtained by both techniques in each land-use were consistent with each other; however, moderately higher net soil erosion estimated by $^{210}$Pb$_{ex}$ at both sites could be a reflection of climate variabilities over the last decades.

Converting forest to vineyard resulted in a significant deterioration in soil quality as was indicated by a significant decline in OM, TN, C:N, CEC, AWC, $f$, and PS, and a significant increase in ECe, pH, BD, and the K factor in both topsoil and subsoil. As a result of deforestation, OM content, which is the most important soil quality indicator, has declined significantly at the vineyard hillslope, leading to a carbon stock loss of about 10.1 and 4.2 Mg ha$^{-1}$ in topsoil and subsoil, respectively. Furthermore, the land-use change weakened the aggregate stability significantly as the PS approximately decreased by half in the vineyard. It also made the vineyard soils susceptible to erosion as K factor surged by 13 and 19% in topsoil and subsoil, respectively.

$^{239+240}$Pu proved to be a valuable tool to quantify the relative contribution of Chernobyl-derived $^{137}$Cs in contaminated areas, which is a prerequisite for applying conversion models with the $^{137}$Cs technique. Obviously, without this knowledge, the results of the $^{137}$Cs method in those areas would not be accurate. Moreover, as found in similar studies, high measurement uncertainties of $^{210}$Pb$_{ex}$ are shown to restrain its application to a great extent. In general, having the highest standard deviation among all measured physicochemical properties, PS is identified to be a powerful tool to study soil aggregate stability and soil quality with low costs. All in all, the results of the present study revealed that deforestation and converting natural vegetation to cropland prompted soil degradation and erosion, deteriorating physicochemical properties of the soil.

## 6. Author contribution

**Maral Khodadadi:** Funding acquisition, Investigation, Resources, Data curation, Software, Writing-Original draft preparation. **Christine Alewell:** Supervision, Writing - Review & Editing. **Mohammad Mirzaei:** Investigation. **Ehssan Ehssan-Malahat:** Investigation. **Farrokh Asadzadeh:** Investigation. **Peter Strauss:** Investigation. **Katrin Meusburger:** Conceptualization, Methodology, Software, Validation, Writing - Review & Editing.



## 7. Competing interests

The authors declare that they have no conflict of interest.

## 8. Acknowledgments

This study has been finalized to support the IAEA Coordinated Research Project (CRP) on "*Nuclear techniques for a better understanding of the impact of climate change on soil erosion in upland agro-ecosystems*" (D1.50.17). The authors would like to thank Nuclear Agriculture Research School, Nuclear Science and Technology Research Institute (NSTRI), Iran.

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





**Table 1: $^{137}$Cs baseline inventory and mean annual precipitation (MAP) in undisturbed locations in different parts of Iran (all values were decay corrected to 1/10/2016).**

| Location | Mean Annual Precipitation (mm) | $^{137}$Cs inventory at reference site (Bq m$^{-2}$) | Reference |
|---|---|---|---|
| Kouhin, centre of Iran | 330 | 1956 ± 107 | Khodadadi et al. (2018) |
| Aghemam Catchment, North-East of Iran | 482 | 2714 | Seyedalipour et al. (2014) |
| Rimeleh catchment, west of Iran | 696 | 1544 | Kalhor (1998); Matinfar et al. (2013) |
| Chaharmahal and Bakhtiari Province, West-South of Iran | 600 | 1730± 32 | Afshar et al. (2010) |
| Gorgan River watershed, North of Iran | 562 | 2178 | Shahoei and Rafahi (1999) |
| Golestan Province, North of Iran | 700-1000 | 3840-4062 | Gharibreza at al. (2020) |
| Zarivar lake watershed, North-West of Iran | 991 | 6152±1266 | This study |




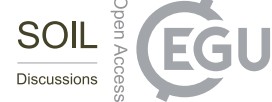

**Table 2: Sediment budget based on $^{137}$Cs and $^{210}$Pb$_{ex}$ dataset at forest and orchard sites.**

| Sediment budget | Forest | | | | | Vineyard | |
| --- | --- | --- | --- | --- | --- | --- | --- |
| | $^{137}$Cs | | $^{210}$Pb$_{ex}$ | | $^{239+240}$Pu | $^{137}$Cs | $^{210}$Pb$_{ex}$ |
| | MODERN | DMM | MODERN | DMM | MODERN | MBM II | |
| **Mean erosion** (Mg ha$^{-1}$ yr$^{-1}$) | 9.8 | 3.3 | 7.0 | 6.3 | 4.1 | 30.9 | 40.3 |
| **Mean deposition** (Mg ha$^{-1}$ yr$^{-1}$) | 7.8 | 1.5 | 2.1 | 3.3 | 2.5 | 10.9 | 23.5 |
| **Gross erosion** (Mg ha$^{-1}$ yr$^{-1}$) | 7.1 | 2.4 | 6.2 | 5.5 | 2.6 | 27.2 | 35.4 |
| **Gross deposition** (Mg ha$^{-1}$ yr$^{-1}$) | 2.1 | 0.4 | 0.3 | 0.4 | 0.9 | 1.3 | 2.9 |
| **Net erosion** (Mg ha$^{-1}$ yr$^{-1}$) | 5.0 | 2.0 | 5.9 | 5.1 | 1.7 | 25.9 | 32.5 |
| **Sediment Delivery Ratio** (%) | 69.9 | 83.0 | 95.9 | 92.8 | 65.5 | 95.2 | 91.9 |





**Table 3: Summary of soil physicochemical properties of the surface and subsurface soils at the forested and orchard hillslope** (the statistical comparison was made between topsoil (0-20 cm) and subsoil (20-40 cm) properties of two land-uses separately). (ECe: Electrical Conductivity, OM: Total Organic Matter, TN: Total Nitrogen, CCE: Carbonate Calcium Equivalent, CEC: Cation Exchange Capacity, BD: Bulk Density, $f$: Porosity, FC: soil water retention at potential equivalent to Field Capacity, PWP: Permanent Wilting Point, AWC: Available Water Content, PS: Percolation Stability, SI: Sealing Index, CI: Crusting Index, and K: soil erodibility).

| Parameter | Topsoil | | Subsoil | |
|---|---|---|---|---|
| | Forest | Vineyard | Forest | Vineyard |
| Clay (%) | $21.04 \pm 3.52^{a*}$ | $26.32 \pm 5.75^{a}$ | $31.37 \pm 1.59^{A**}$ | $23.35 \pm 2.66^{B}$ |
| Silt (%) | $53.96 \pm 4.01^{a}$ | $50.29 \pm 4.26^{a}$ | $51.09 \pm 7.39^{A}$ | $53.22 \pm 3.79^{A}$ |
| Sand (%) | $25.00 \pm 4.12^{a}$ | $23.39 \pm 1.92^{a}$ | $17.54 \pm 5.80^{A}$ | $23.43 \pm 1.12^{A}$ |
| Gravel (%) | $18.32 \pm 6.74^{a}$ | $13.77 \pm 5.51^{a}$ | $24.34 \pm 9.34^{A}$ | $27.22 \pm 6.95^{A}$ |
| pH | $7.17 \pm 0.15^{b}$ | $7.58 \pm 0.33^{a}$ | $7.25 \pm 0.11^{B}$ | $7.49 \pm 0.17^{A}$ |
| ECe (dS m$^{-1}$) | $1.95 \pm 0.31^{b}$ | $2.40 \pm 0.20^{a}$ | $1.56 \pm 0.10^{A}$ | $1.45 \pm 0.16^{A}$ |
| OM (%) | $2.83 \pm 0.30^{a}$ | $1.44 \pm 0.26^{b}$ | $1.72 \pm 0.13^{A}$ | $1.16 \pm 0.23^{B}$ |
| TN (gr kg$^{-1}$) | $0.20 \pm 0.08^{a}$ | $0.16 \pm 0.02^{b}$ | $0.16 \pm 0.06^{A}$ | $0.12 \pm 0.02^{B}$ |
| C:N | $72.89 \pm 20.33^{a}$ | $42.21 \pm 4.05^{b}$ | $61.18 \pm 22.18^{A}$ | $50.55 \pm 12.69^{A}$ |
| CCE (%) | $2.79 \pm 0.86^{a}$ | $3.25 \pm 0.33^{a}$ | $3.42 \pm 0.28^{A}$ | $3.78 \pm 0.52^{A}$ |
| CEC (cmol$^+$ kg$^{-1}$) | $27.91 \pm 0.82^{a}$ | $22.46 \pm 0.94^{b}$ | $34.86 \pm 1.59^{A}$ | $24.64 \pm 0.20^{B}$ |
| BD (Mg m$^{-3}$) | $1.09 \pm 0.56^{b}$ | $1.37 \pm 0.74^{a}$ | $1.28 \pm 0.72^{B}$ | $1.55 \pm 0.95^{A}$ |
| $f$ (%) | $59.03 \pm 2.11^{a}$ | $48.36 \pm 2.78^{b}$ | $51.89 \pm 2.73^{A}$ | $41.38 \pm 3.60^{B}$ |
| FC (%) | $38.58 \pm 1.33^{a}$ | $32.81 \pm 1.60^{b}$ | $32.38 \pm 1.10^{A}$ | $30.76 \pm 1.21^{B}$ |
| PWP (%) | $15.57 \pm 1.87^{a}$ | $14.91 \pm 2.08^{a}$ | $14.42 \pm 1.32^{B}$ | $14.41 \pm 1.59^{A}$ |
| AWC (%) | $23.01 \pm 2.93^{a}$ | $17.90 \pm 1.35^{b}$ | $17.96 \pm 1.65^{A}$ | $16.35 \pm 1.49^{B}$ |
| PS (gr H$_2$O 600s$^{-1}$) | $309.35 \pm 43.61^{a}$ | $137.92 \pm 66.28^{b}$ | $160.45 \pm 40.96^{A}$ | $101.18 \pm 62.85^{B}$ |
| SI (%) | $3.84 \pm 0.57^{a}$ | $1.88 \pm 0.34^{b}$ | $2.15 \pm 0.11^{A}$ | $1.51 \pm 0.29^{B}$ |
| CI | $1.04 \pm 0.05^{b}$ | $1.27 \pm 0.11^{a}$ | $1.16 \pm 0.08^{B}$ | $1.53 \pm 0.07^{A}$ |
| K (t ha hr MJ$^{-1}$ ha mm$^{-1}$) | $0.044 \pm 0.005^{b}$ | $0.051 \pm 0.009^{a}$ | $0.055 \pm 0.005^{B}$ | $0.067 \pm 0.005^{A}$ |

*Different lowercase letters indicate significant differences at the p-level<0.05 in topsoil properties of two land-uses.

**Different capital letters indicate significant differences at the p-level<0.05 in subsoil properties of two land-uses.




**Table 4: $^{239+240}$Pu baseline inventory and mean annual precipitation (MAP) in undisturbed locations.**

| Location | Mean Annual Precipitation (mm) | $^{239+240}$Pu inventory at reference site (Bq m$^{-2}$) | Reference |
|---|---|---|---|
| Australia | 1200 | 8.8 | Tims et al. (2013) |
| China | 600 - 800<br>800 - 1000<br>235 - 238 | 86.9±3.1<br>44.9 - 54.6 (in Hubei)<br>32.4 (in Lanzhou) | Xu et al. (2013)<br>Zheng et al. (2009)<br>Dong et al. (2010) |
| Europe | 500-3500 | 8.0 to 380.2 (53.3) | Meusburger et al. (2020) |
| Germany | 968 | 59±8 | Schimmack et al. (2001) |
| South Korea | -<br>-<br>1599 | 18.4 (in Sesan);<br>101.8 (in Euiwang)<br>55 ±8 | Lee et al. (1996)<br>Lee et al. (1996)<br>Meusburger et al. (2016) |
| Switzerland | 1400 | 67±13 and 83±11 | Meusburger et al. (2018); Schaub et al. (2010); Alewell et al. (2014) |




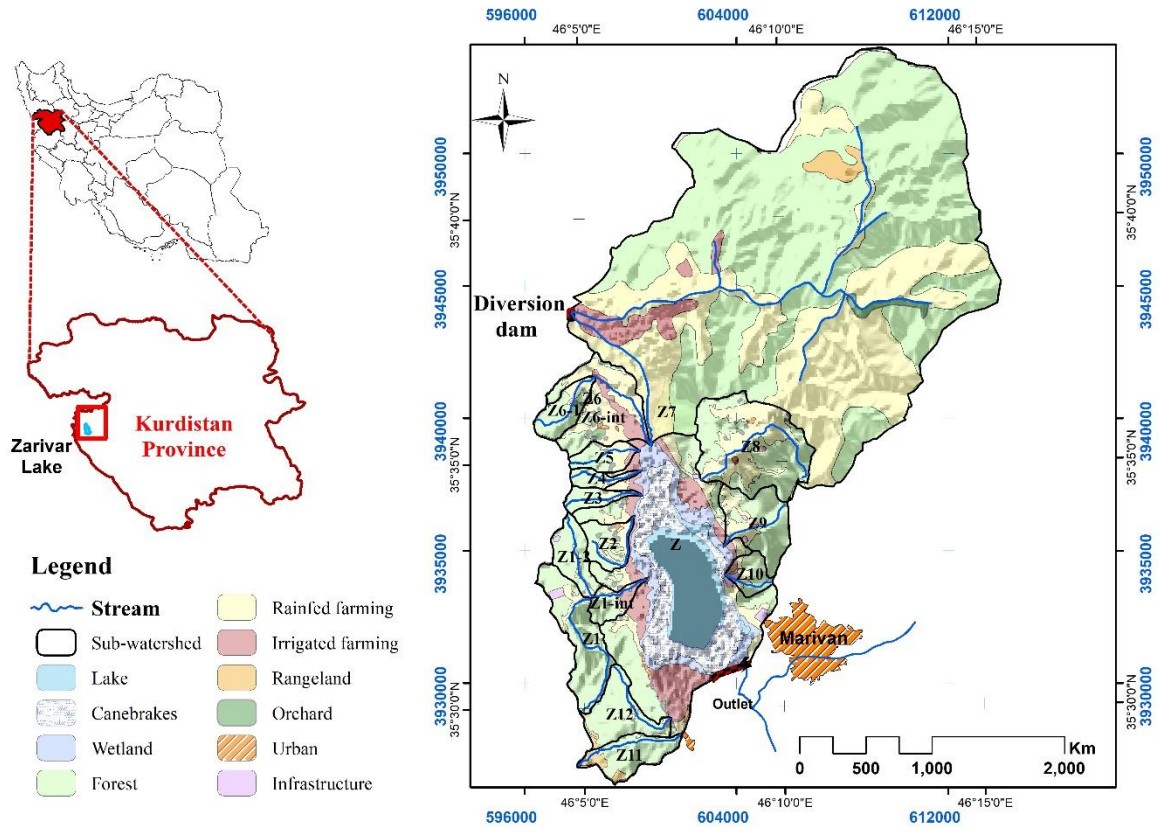

**Figure 1: The location of the Zarivar Lake watershed in Kurdistan Province, Iran.**



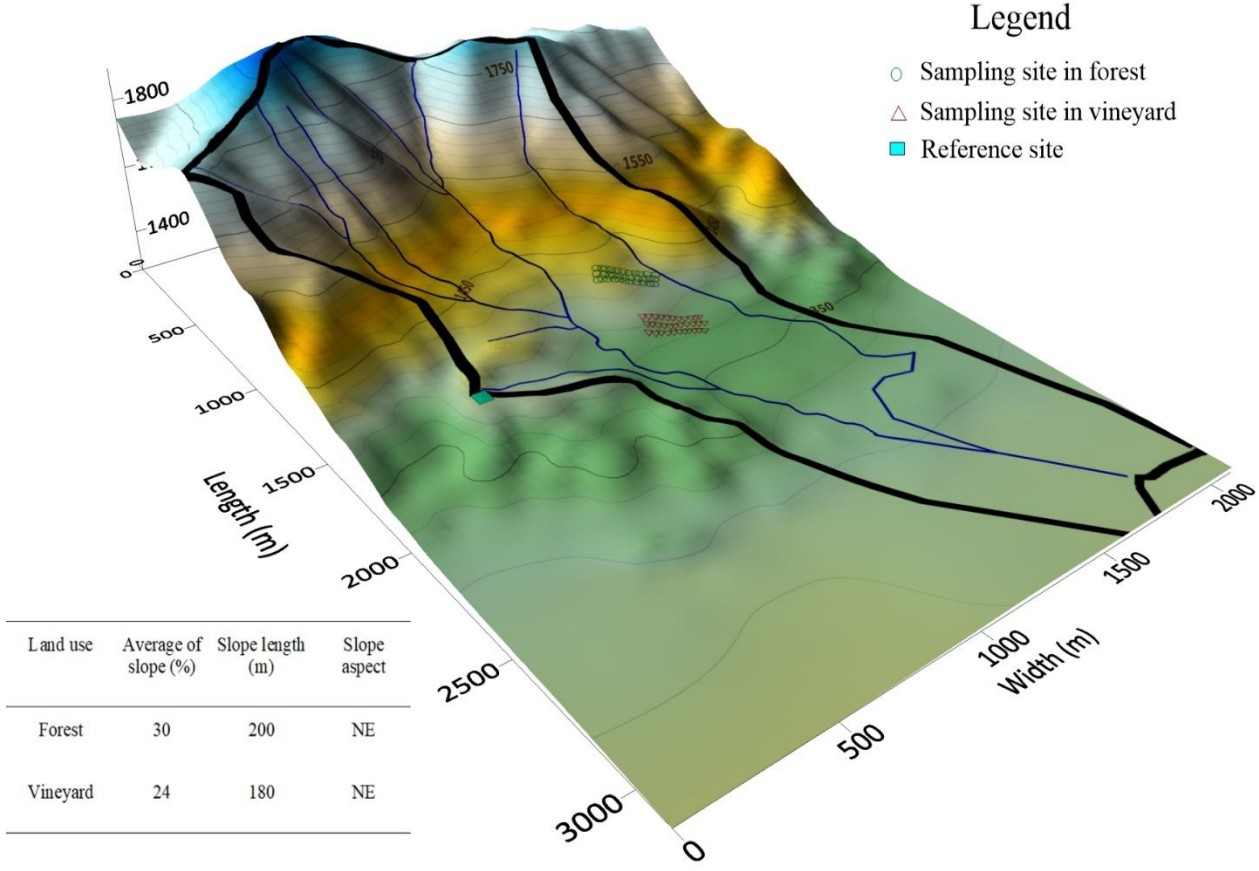


**Figure 2: The location of the reference site and the study hillslopes at Z3 sub-watershed.**





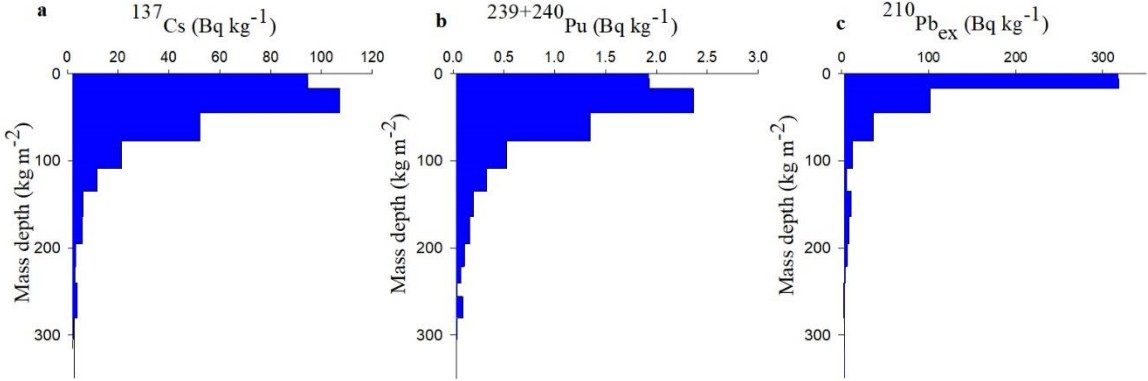

**Figure 3: Depth distribution of** [137]**Cs (a),** [210]**Pb$_{ex}$ (b) and** [239+240]**Pu at the reference site.**


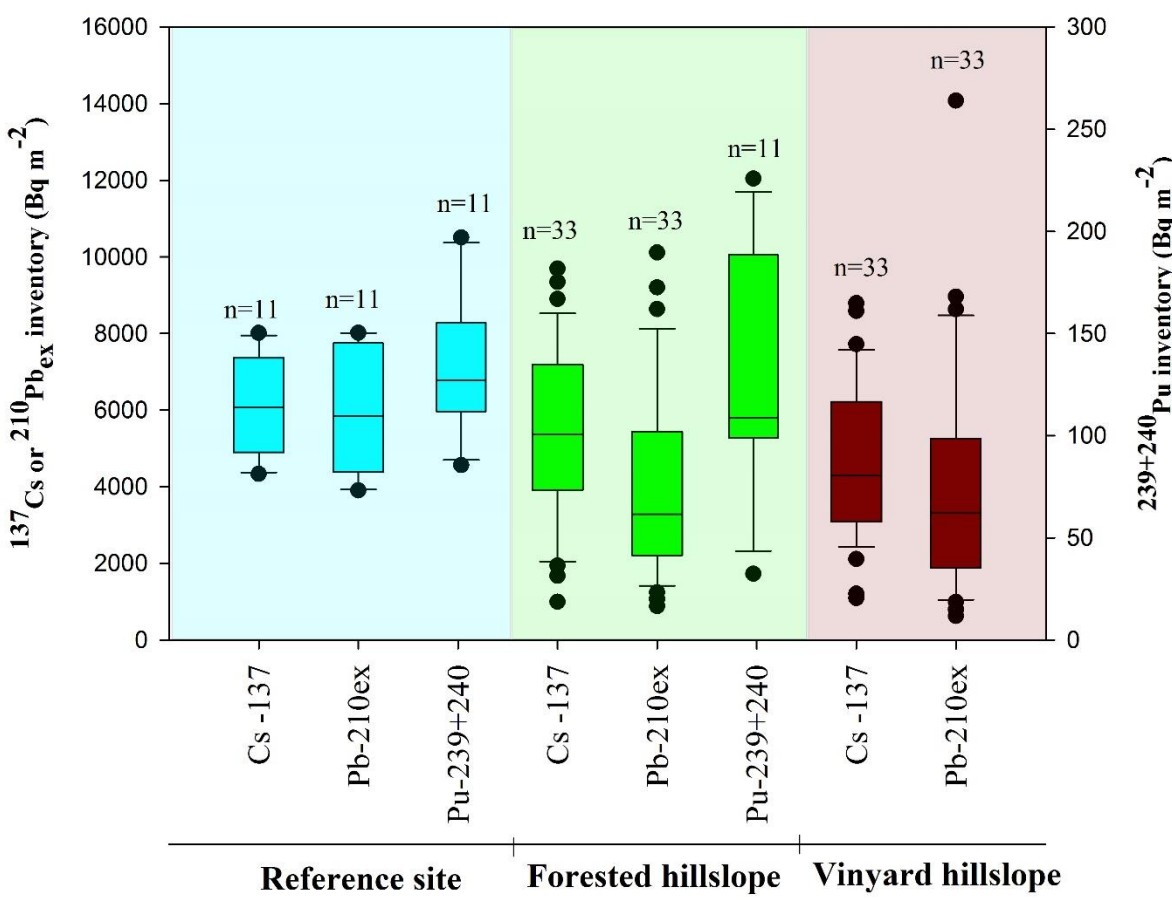

**Figure 4:** $^{137}$Cs, $^{210}$Pb$_{ex}$, and $^{239+240}$Pu inventories at reference, forest, and vineyard sites (the scale of Y axes for $^{137}$Cs and $^{210}$Pb$_{ex}$ is different from that of $^{239+240}$Pu).






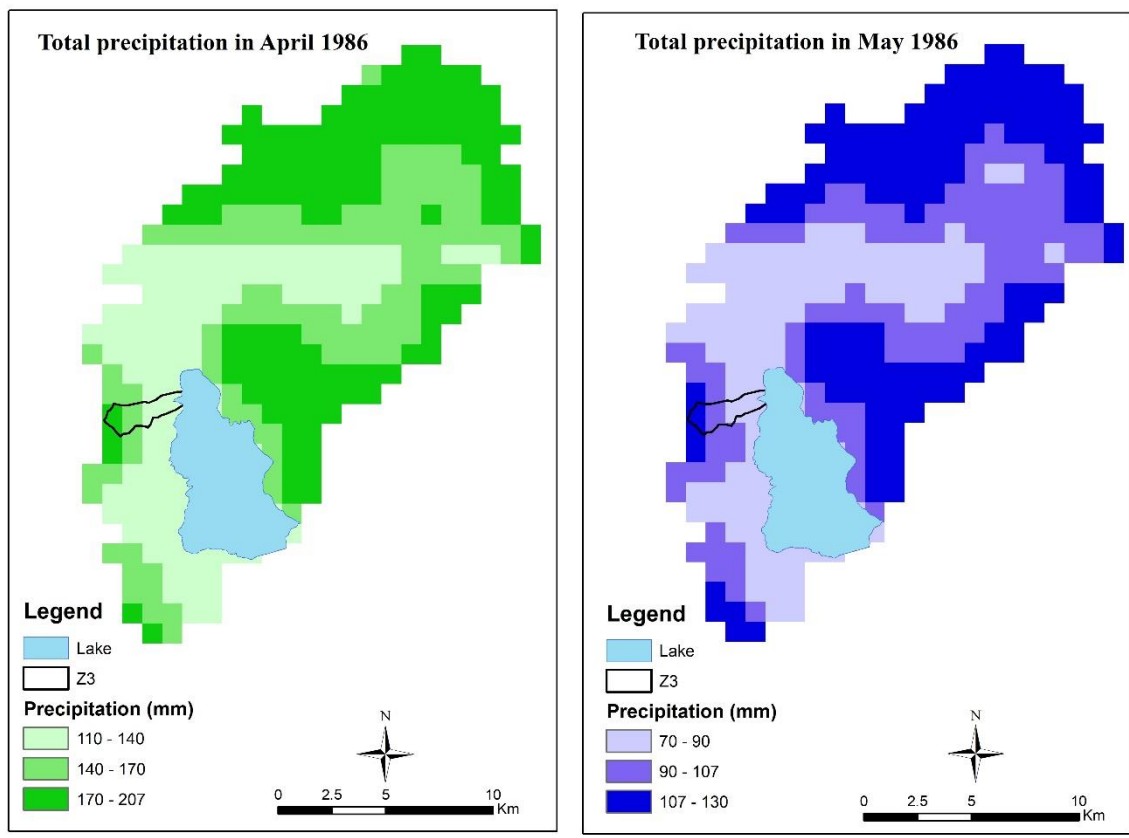

Figure 5: Total precipitation in April and May 1986 (from CHELSA dataset; Climatologies at high resolution for the earth's land surface areas).



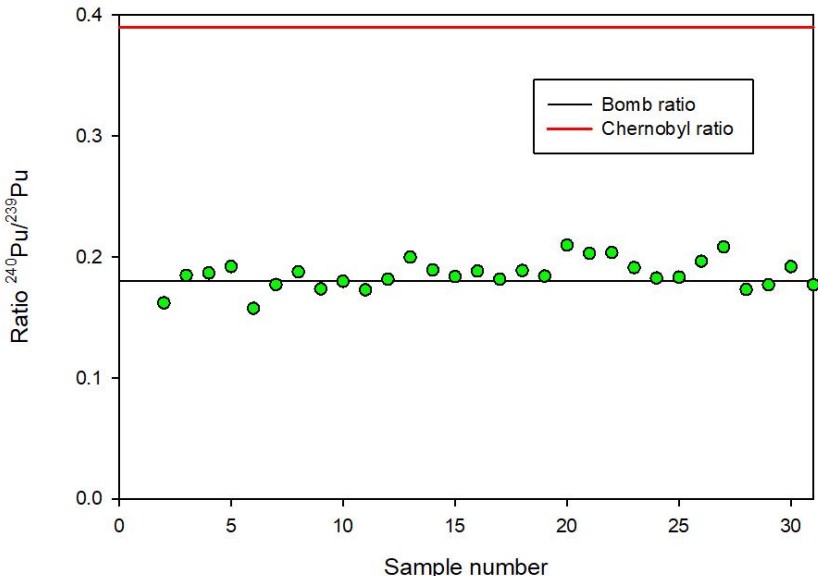

**Figure 6: Ratio of $^{240}$Pu/$^{239}$Pu of all samples in the forested hillslope.**

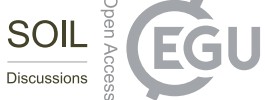

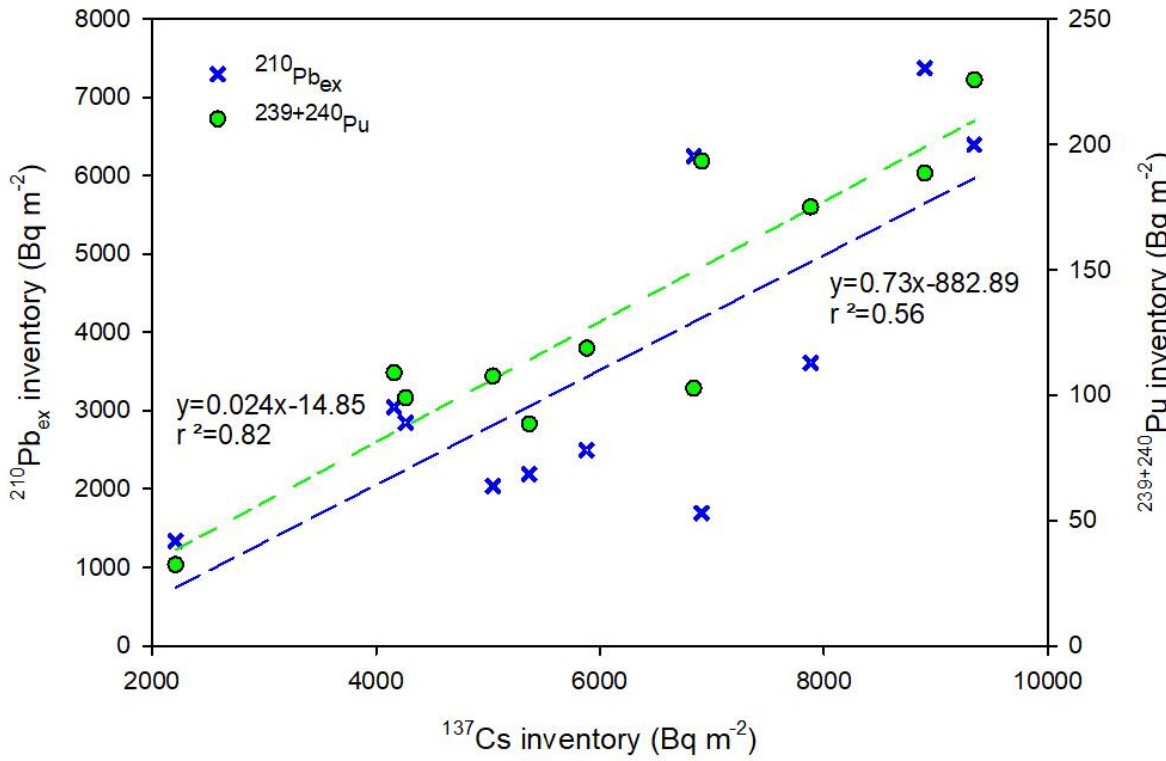

**Figure 7: Comparison of $^{210}Pb_{ex}$ and $^{239+240}Pu$ with $^{137}Cs$ inventories along a transect in the forested hillslope.**





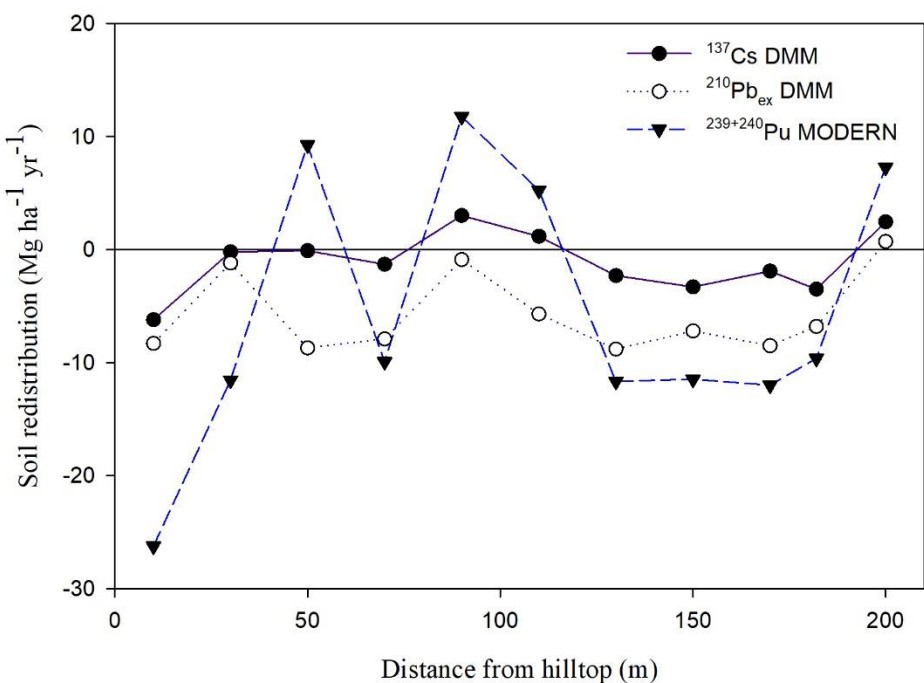

**Figure 8: FRNs derived soil redistribution rates in a forested transect using MODERN and Diffusion and Migration model (DMM) (negative values indicate erosion, whereas positive values indicate deposition).**



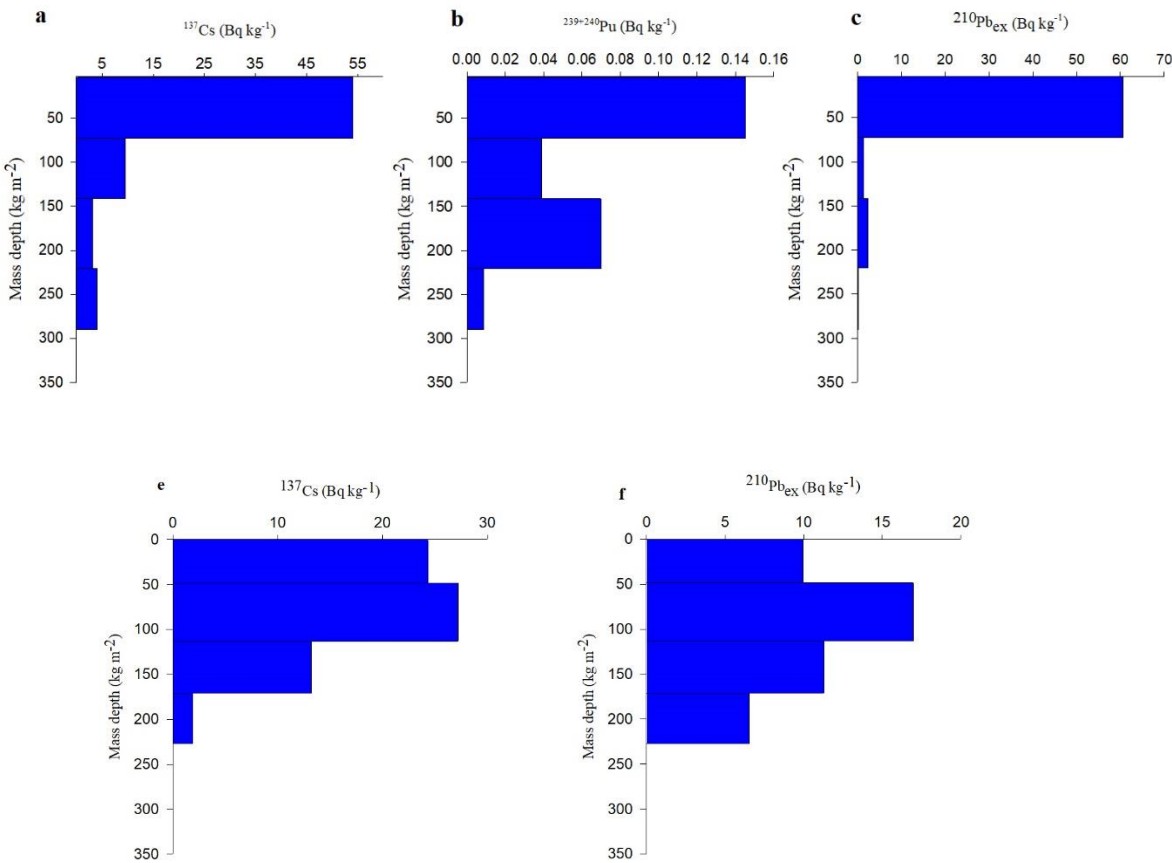

**Figure 9: Depth distribution of** [137]**Cs (a),** [210]**Pb$_{ex}$ (b) and** [239+240]**Pu (c) in an erosional site in forested hillslope and** [137]**Cs (e) and** [210]**Pb$_{ex}$ (f) at an erosional site in vineyard hillslope.**