# Peer review of "Deforestation effects on soil erosion rates and soil physicochemical properties in Iran: a case study of using fallout radionuclides in a Chernobyl contaminated area"

_SOIL, 2021_

## Referee Comment (RC1)

General comments

The manuscript is based on the results of assessing the erosion/deposition rates using FRN as markers. The following questions arise when reading the manuscript:

1/ Based on Figure 4, we can conclude that the variability of 137Cs on a forested slope is higher than on vineyards. This clearly indicates that the Chernobyl fallout was extremely uneven in the area. It is obvious, that the initial 137Cs spatial variability of Chernobyl fallout too high for using the 137Cs technique for evaluating the soil losses based only on one reference location. It is necessary to evaluate 137Cs initial inventory minimum at three reference locations for the evaluation of the trend of initial fallout (See:  Handbook for the Assessment of Soil Erosion and Sedimentation Using Environmental Radionuclides, F.Zapata for details). So authors aren't able to confirm the correctness of their evaluation of soil losses, based on Chernobyl-derived 137Cs because they have only one reference location on the distance of about 1 km from the studied site.

2/ On the other hand, if the mean 210Pb inventories values are approximately the same for the forested slope and vineyard (see bottom of page 9:  210Pbex inventories at 4068.3±2345.8 and 3990.1±2892.2 Bq m-2, respectively for forested and vineyard hillslopes) , then it is completely incomprehensible how such a large difference in net erosion rates (calculated based on  210Pbex) between the forested area and the vineyards was received (see Table 2). It is quite obvious that this is a very gross error in the calculations.

3/ The local spatial variation of the 137Cs fallout, as well as the other radionuclides, is >20% on the reference location. Isn't recommended to use FRD for evaluation of soil loss/gain in areas where initial fallout variability > 20% due to very high uncertainty of the results (see papers written by D. Walling and the other experts in the application of FNR for the evaluation of soil redistribution rates).

4/ Authors indicate that …The soil redistribution rates were, therefore, estimated using the Diffusion and Migration Model 185 (DMM) (Walling et al., 2002, 2014) in forested hillslope, Mass Balance Model II (MBM II) (Walling et al., 2002, 2014) in cultivated hillslope, and Modelling Deposition and Erosion rates with RadioNuclides (MODERN; Arata et al., 2016a, 2016b) – It is necessary to present the equations for all conversion models and to explain how you determine the parameters for each model.

5/ In addition application FRN for evaluation soil erosion rates in the forest isn't possible at all due to the influence of the crown of trees on the initial spatial variability.

Specific comments

1/ Are you sure that sampling only 40 cm layer is enough for determination of FRN total inventory in deposition location? How you can confirm that?

2/ In the Supplementary Material, the 137Cs depth distribution in the lake is presented. But it is completely incomprehensible how it was obtained?

3/ Figure 5 – how do you construct both maps with so high spatial resolution? Why was the total precipitation in April and May used to construct these maps? The accident at the Chernobyl nuclear power plant occurred on April 26 and the bulk of the Chernobyl fallout was observed until May 15 and was associated with the fallout of only one rain at a distance from Chernobyl.

4/ In the references, there are practically no papers on the use of 137 for assessing the erosion/sedimentation rates, prepared on the basis of research in the Chernobyl-affected areas (the UK, Poland, Belarus, Russia, Ukraine,  Scandinavia, the Baltic States)

5/ It is not specified anywhere in the ms when the forest was cut down and vineyards were planted on the studied site.

Technical corrections:

1/ Introduction … than five times, increasing from 2.6 million ha (Wilber, 1948) to 18.5 million ha – should be more than seven times

---

## Author Comment (AC1)

**Answer to Reviewer#1**

*General comments*

*The manuscript is based on the results of assessing the erosion/deposition rates using FRN as markers. The following questions arise when reading the manuscript:*

*1/ Based on Figure 4, we can conclude that the variability of 137Cs on a forested slope is higher than on vineyards. This clearly indicates that the Chernobyl fallout was extremely uneven in the area. It is obvious, that the initial 137Cs spatial variability of Chernobyl fallout too high for using the 137Cs technique for evaluating the soil losses based only on one reference location. It is necessary to evaluate 137Cs initial inventory minimum at three reference locations for the evaluation of the trend of initial fallout (See: Handbook for the Assessment of Soil Erosion and Sedimentation Using Environmental Radionuclides, F.Zapata for details). So authors aren't able to confirm the correctness of their evaluation of soil losses, based on Chernobyl-derived 137Cs because they have only one reference location on the distance of about 1 km from the studied site.*

At the study site, the variability of the 137Cs baseline reference inventory was estimated at 21% (11 samples). This is not an exceptionally high value and there, in fact, were many studies applying FRNs in the areas with a higher spatial variability of 20% e.g., Nagle et al. (2000), Sutherland (1991), Meusburger et al. (2016), Teramage et al. (2015).

The spatial variability of 137Cs inventories in forests is high (expressed by a CV of 41.3); however, it was equal to vineyard site (see table below and 4.1.2. Fallout radionuclides inventories and soil redistribution rates at slope transects section in the MS line 255). The variability in the forest may be the result of initial fallout heterogeneity but also from actual soil redistribution at the steep slope. Thus, from the heterogeneity in an erosional site it cannot be deduced whether the 137Cs technique is applicable or not.

Table- Average, standard deviation (SD), and coefficient of variation (CV) of 137Cs inventories in two hillslopes

| Land-use | Average (Bq m$^{-2}$) | SD (Bq m$^{-2}$) | CV (%) |
|---|---|---|---|
| Forest | 5389.94 | 2227.97 | 41.34 |
| Vineyard | 4646.61 | 1921.82 | 41.35 |

*2/ On the other hand, if the mean 210Pb inventories values ââare approximately the same for the forested slope and vineyard (see bottom of page 9: 210Pbex inventories at 4068.3±2345.8 and 3990.1±2892.2 Bq m-2, respectively for forested and vineyard hillslopes), then it is completely incomprehensible how such a large difference in net erosion rates (calculated based on 210Pbex) between the forested area and the vineyards was received (see Table 2). It is quite obvious that this is a very gross error in the calculations.*

Both land use types are subject to very different cultivations: the vineyards are regularly ploughed, while the forest is not. In fact, similar inventory values in the upper most soil represent very different erosion rates. At the vineyard, deposited FRN will be mixed by ploughing into deeper soil layers, down to a depth of 30 cm. Thus, different conversion models should be applied at the uncultivated and cultivated sites. As asserted by F. Zapata in his book "*Handbook for the Assessment of Soil Erosion and Sedimentation Using Environmental Radionuclides*," since FRNs are mixed and diluted in tillage depth at the cultivated sites, its loss in cultivated lands corresponds with a higher amount of soil loss compared to uncultivated lands

in which FRNs are concentrated in the upper parts of the soil profile (Zapata, 2002). Therefore, the conversion models developed for cultivated lands consider the above-mentioned fact and predict higher amount of soil loss/gain for the similar amounts of FRNs in cultivated lands. Thus, the difference in erosion rates is not due to a calculation error.

 *3/ The local spatial variation of the 137Cs fallout, as well as the other radionuclides, is >20% on the reference location. Isn't recommended to use FRD for evaluation of soil loss/gain in areas where initial fallout variability > 20% due to very high uncertainty of the results (see papers written by D. Walling and the other experts in the application of FRN for the evaluation of soil redistribution rates).*

137Cs, 239+240Pu, and 210Pbex variability at the reference site were 21, 23, and 25% respectively; however, there are many examples of FRNs usage in areas with relatively higher CVs in the literature (see the answer to question 1). Furthermore, IAEA (2014) suggested that if the coefficient of variation (CV) of the FRN inventory at a specific reference site exceeds 30%, this is an indication of its unsuitability and reference sample numbers should be increased.

*- IAEA (2014), Guidelines for using fallout radionuclides to* assess *erosion and effectiveness of soil conservation strategies, International Atomic Energy Agency, IAEA-TECDOC-1741. Vienna, Austria, 1-213, 2014a.*

"Page:39: If the standard deviation obtained for this sample set proves too high (CV% > 30%), more samples should be taken (Mabit et al., 2014)". So, our CVs are clearly lower than this value. Therefore, the sentence (in line:364) was modified as follows:

*-Mabit, L., Chhem-Kieth, S., Dornhofer, P., Toloza, A., Benmansour, M., Bernard, C., Fulajtar, E. and Walling, D.E., 2014. 137 Cs: A Widely Used and Validated Medium Term Soil Tracer. IAEA TECDOC SERIES, p.27.*

"Different percentages have been reported in the literature for the allowable CV of FRNs at the reference site; for instance, Sutherland (1996) and Mabit et al. (2012) suggested it to be 20% at 90% level of confidence, while in IAEA (2014) the figure was declared to be below 30%. The selected reference site here met the criteria of the FRNs' CVs, i.e. below 30%."

 *4/ Authors indicate that ...The soil redistribution rates were, therefore, estimated using the Diffusion and Migration Model 185 (DMM) (Walling et al., 2002, 2014) in forested hillslope, Mass Balance Model II (MBM II) (Walling et al., 2002, 2014) in cultivated hillslope, and Modelling Deposition and Erosion rates with RadioNuclides (MODERN; Arata et al., 2016a, 2016b) – It is necessary to present the equations for all conversion models and to explain how you determine the parameters for each model.*

As suggested, the parameters for each model have been added to the MS (see below). Since the conversion models have been described in numerous publications, we prefer to refer to these studies instead of providing the equations to keep this manuscript as concise as possible. However, if you or the editor prefer to have this information, we could provide it in the supplementary material.

We thus will change the paragraph to:

"For $^{137}$Cs and $^{210}$Pb$_{ex}$ the MODERN and the DMM were applied at the forested site, while at the vineyard site, the MBM II (i.e. without tillage component) was used (Table 2). Particle size factor was taken to be 1 for all models. For DMM, the migration rate (V), the diffusion rate (D), and the relaxation depth (H) were determined using the depth profile of $^{137}$Cs and $^{210}$Pb$_{ex}$ at the reference site.  For MBM II, a value of 4 kg m$^{-2}$ for H and a value of 0.5 for proportion factor ($\gamma$) were set (Walling et al., 2002; 2014). To estimate the soil redistribution rate at each sampling point, the Excel Add-in Bradiocalc.xla (see http://www-naweb.iaea.org/nafa/swmn/models-tool-kits.html) was used for executing the DMM and MBM models. For more details on equations of the models and definitions of their parameters, readers are

referred to Walling et al. (2002; 2014). Furthermore, the MATLAB code (https://duw.unibas.ch/de/umweltgeowissenschaften/forschung-fg-alewell/modern/registration/) was applied for executing MODERN, find more details on the model and its parameters in Arata et al. (2016a; 2016b)."

*5/ In addition application FRN for evaluation soil erosion rates in the forest isn't possible at all due to the influence of the crown of trees on the initial spatial variability.*

We totally agree with you, that due to the influence of interception deposition, canopy throughfall, stemflow and general canopy heterogeneity the interpretation of the data has to be done with caution. However, as the CV of FRN was similar in forests as it was in vineyards, we do think it is acceptable to apply this method to our forested areas. As reported in the MS, there are many studies in which FRNs have been used in the forested areas worldwide (e.g. Gaspar et al., 2013; Wakiyama et al., 2010; Gharibreza et al., 2013; 2020). Commonly, it is proposed to collect samples as far away from the tree trunks as possible to avoid the influence of stem flow. We followed this advice. Furthermore, we collected >30 samples within the forested site, enabling us to successfully gain such comparable CVs between forests and vineyards and average out some of the small-scale variability caused by the throughfall pattern.

*Specific comments*

*1/ Are you sure that sampling only 40 cm layer is enough for determination of FRN total inventory in deposition location? How you can confirm that?*

Thank you for this comment and sorry to be imprecise. We added this statement to the MS: "in locations where deposition was expected, whenever the soil was deep enough, the samples were collected up to 40 and 50 cm in forest and vineyard hillslopes, respectively."

2/ In the Supplementary Material, the 137Cs depth distribution in the lake is presented. But it is completely incomprehensible how it was obtained?

 The presented graph was meant to confirm whether or not the study site was affected by Chernobyl (according to the number of the peaks); sorry for the confusion, we will not present the lake core data in this manuscript.

*3/ Figure 5 – how do you construct both maps with so high spatial resolution? Why was the total precipitation in April and May used to construct these maps? The accident at the Chernobyl nuclear power plant occurred on April 26 and the bulk of the Chernobyl fallout was observed until May 15 and was associated with the fallout of only one rain at a distance from Chernobyl.*

 We are sorry, but these monthly datasets had to be used since no local daily precipitation data was recorded in the site at the time of the expected Chernobyl fallout. The graphs presented in the MS have the original resolution provided by the data source mentioned in the MS (https://chelsa-climate.org/).

As elaborated in the MS and in other studies (Meusburger et al., 2020), there is a significant correlation between mean annual precipitation and the FRNs inventories. We also showed the maps to evaluate whether there was a heterogeneous rainfall distribution in the time frame of the possible Chernobyl fallout.

*4/ In the references, there are practically no papers on the use of 137 for assessing the erosion/sedimentation rates, prepared on the basis of research in the Chernobyl-affected areas (the UK, Poland, Belarus, Russia, Ukraine, Scandinavia, the Baltic States).*

We cited several studies that have been conducted in areas with Chernobyl-derived 137Cs, e.g. in Germany (Schimmack et al., 2001; 2002), Switzerland (Alewell et al., 2014; Meusburger et al., 2016; 2018), and Austria (added in the revised version, Meusburger et al., 2020). Switzerland and Austria experienced very high Chernobyl contributions as compared to the global fallout. As our study is not a review, it is beyond the scope of this paper to consider all publications on this topic. However, we followed your advice and included some research in the areas closest to our study site e.g. Gharibreza et al. (2021) and Vahabi-Moghaddam and Khoshbinfar (2012) (following the advice of Reviewer #3 as well).

5/ It is not specified anywhere in the ms when the forest was cut down and vineyards were planted on the studied site.

Sorry, this information was placed in the wrong section (i.e. 2.3 Soil sampling and soil physicochemical properties line:189) and now is mentioned in the appropriate paragraph (i.e. 2.1. Study area).

Technical corrections:

1/ Introduction … than five times, increasing from 2.6 million ha (Wilber, 1948) to 18.5 million ha – should be more than seven times

Profound thanks, this is a mistake. We did correct it in the revised MS.

---

## Author Comment (AC2)

**Answer to Reviewer#2,**

General comments

Overall, the study focuses on a relevant topic and presents interesting results and is therefore definitely worth publishing after some modifications and improvements to the current manuscript have been made. This manuscript represents valuable results, and a good approximation to better understand the differences and similarities between different methods to calculate soil redistribution rates and its relationship with some physical properties and soil nutrients.

The study includes interesting findings first reported in these areas of Iran and fits within SOIL scopes. However, in its current state, I think it requires quite a bit more work in terms of rearranging and editing before it is ready for publication. Overall, the study focuses on a relevant topic and presents interesting, results and is therefore worth publishing after some major modifications and improvements to the current manuscript have been made.

Thank you for your appreciation and your positive feedback. We highly appreciate this.

First, I think authors should clarify some of the previous reviewer questions. To avoid some repetition, I have been more focused on specific details. Apart from the general comments about each section, I have included some comments about the figures (14) and the text (17). An additional major issue I see in this manuscript is the enormous number of references included. If I am not wrong, there are more than 100 references. I think it is more than needed for a novel study.

Thank you for all your work. Please see our detailed answer to reviewer 1 and to reviewer 3. We totally agree that 100 references are too many. We thus deleted those, we think, were not absolutely necessary.

Introduction

I think you need to emphasise this work's aims at the end of this section, the authors present an interesting study, but sometimes it is easy to get confused because the aim of the manuscript is repeated trough the introduction section. Thus, it could be useful to be more specific at the end of this section and reorder or rewrite some parts of the introduction.

As suggested, we restructured the introduction in a way that the study site and its related challenges have moved to the last paragraph of the introduction.

Results and discussion

In this section, I would encourage the authors to synthesise the results and focus only on the results. Sometimes it seems that the discussion is a little bit mixed with the results.

Furthermore, I think more than 2800 words for the discussion section is excessive. In this section, authors tend to repeat most of the results, and there is an excessive number of comparisons with other studies. I think the manuscript could be reduced here. This statement is quite clear because authors have included more than 100 references, what it is just disproportionate. Don't you think? For example, 50-60 references are considered a high number of references, still acceptable but high.

As suggested, we tried to separate the results and discussion as much as possible. In total, we reduced the section by 170 words. We agree with Reviewer 2 about the number of the references being too many as well; therefore, we deleted less relevant references.

Conclusions

I think that the text could be improved here (to avoid restricting it to a repetition of what was written before). The first 6 lines did not include a conclusion and just repeated previous parts of the manuscript. After a very rapid summary of the main results, you could stress the potential novel avenues for research in the future.

As suggested by the Reviewer # 2, we added the potential novel avenues for research in the future and rewrote the conclusion as follows (the number of words reduced from 717 to 589):

"Forests and forest soils store carbon and therefore play a decisive role in mitigating climate change impacts. In Iran, over the past decades, a rapidly growing population has induced an increasing demand for food, so one of the most rapid land-use changes, i.e. conversion of land under natural vegetation into arable lands, has been occurring. The present study was undertaken to quantify the impact of deforestation on soil loss using fallout radionuclides and soil physicochemical properties.

The deposition of the $^{137}$Cs fallout from different sources (Chernobyl fallout vs. global fallout) could be determined via $^{239+240}$Pu isotopes since Pu isotopes, as confirmed by $^{240}$Pu/$^{239}$Pu atom ratios, originated exclusively from the global fallout. From the $^{137}$Cs/$^{239+240}$Pu ratio, it was evident that half of the $^{137}$Cs found in the site was Chernobyl-derived. The mean reference inventory of $^{137}$Cs at 6152±1266 Bq m$^{-2}$ was higher than previous reports in different parts of Iran. The reasons behind that can be the site being closer to the Chernobyl site than other studied sites in Iran, a higher mean annual precipitation, and relatively high rainfall after the Chernobyl incident. The mean reference inventories of $^{210}$Pb$_{ex}$ estimated at 6079±1511 Bq m$^{-2}$ was in accord with the reported value in the central part of Iran. The mean reference inventory of $^{239+240}$Pu at 135 ±31 Bq m$^{-2}$ was higher than values reported in other parts of the globe, which might be attributed to high initial bomb-derived deposition in the study site during 1953–1964. Nevertheless, measurements from other parts of the country are required to confirm the hypothesis of a high initial bomb-derived Pu fallout. $^{239+240}$Pu proved to be a valuable tool to quantify the relative contribution of Chernobyl-derived $^{137}$Cs in contaminated areas, which is a prerequisite for applying conversion models with the $^{137}$Cs technique.

Both $^{137}$Cs and $^{210}$Pb$_{ex}$ radionuclides indicated that deforestation has increased annual soil loss by about five times. Notable is that the values obtained by both techniques in each land-use were consistent. Moderately higher net soil erosion rates were estimated by $^{210}$Pb$_{ex}$ at both sites. $^{210}$Pb$_{ex}$ is more sensitive to recent erosion events (past 20 years). However, the uncertainties associated with this FRN were high. Thus, the conclusion that erosion increased based on these results cannot be drawn and needs further investigation.

As a result of deforestation, OM content, which is the most important soil quality indicator, has declined significantly at the vineyard hillslope, leading to a carbon stock loss of about 10.1 and 4.2 Mg ha$^{-1}$ in topsoil and subsoil, respectively. Furthermore, the land-use change significantly weakened aggregate stability as the PS approximately decreased by half in the vineyard. PS is identified to be a powerful tool to study soil aggregate stability and soil quality with low costs. Thus, converting forests to vineyards resulted in a significant deterioration in soil quality which will likely impact the soil productivity and food security.

All in all, the results of the present study revealed that deforestation and converting natural vegetation to cropland prompted soil degradation and erosion, deteriorating physicochemical properties of the soil. However, further measurements of $^{239+240}$Pu in the region are still required to understand the spatial variability of relative contribution of Chernobyl-derived $^{137}$Cs. This would assist with producing more

precise maps of spatial distribution of Chernobyl-derived $^{137}$Cs input. Further investigations are also necessary to verify if PS is a suitable soil quality indicator in other areas. Furthermore, in order to better understand the impact of deforestation on soil properties and soil loss, it is suggested that they be monitored at different time windows after deforestation."

497 you could specify the location without using brackets, please rewrite the sentence.

 The sentence has been modified as follows:

"The reasons behind that can be the site being closer to the Chernobyl site than other studied sites in Iran, a higher mean annual precipitation, and relatively high rainfall after the Chernobyl incident."

496-500 Following table 1, the study pursued in Golestan Province with similar mean annual precipitation (MAP), showed significantly lower values than your study. Do authors think that all the difference could be due to the location even if both are located in the northern part of Iran?

 The distance between Golestan (located in Northern east of Iran) and our site (in Northern west of Iran) is around 850 km which is almost half of the distance between our study site and Chernobyl (which is 2000 km). Also, we added two new references i.e. the Gharibreza et al. (2021) and Vahabi-Moghaddam and Khoshbinfar (2012) whose study sites are located in the north of Iran and whose results are entirely compatible with ours (see below).

Table 1: 137Cs baseline inventory and mean annual precipitation (MAP) in undisturbed locations in different parts of Iran (all values were decay corrected to 1/10/2016).

| Location | Mean Annual Precipitation (mm) | 137Cs inventory at reference site (Bq m-2) | Reference |
|---|---|---|---|
| Kouhin, centre of Iran | 330 | 1956 ± 107 | Khodadadi et al. (2018) |
| Aghemam Catchment, North-East of Iran | 482 | 2714 | Seyedalipour et al. (2014) |
| Rimeleh catchment, west of Iran | 696 | 1544 | Kalhor (1998); Matinfar et al. (2013) |

| | | | |
|---|---|---|---|
| Chaharmahal and Bakhtiari Province, West-South of Iran | 600 | 1730± 32 | Afshar et al. (2010) |
| Gorgan River watershed, North of Iran | 562 | 2178 | Shahoei and Rafahi (1999) |
| Golestan Province, North of Iran | 700-1000 | 3840-4062 | Gharibreza at al. (2020) |
| | 1000 | 3570 -5270 | Vahabi-Moghaddam and Khoshbinfar (2012) |
| Gilan Province, North of Iran | 1209 | 6180 | Gharibreza et al. (2021) |
| Zarivar lake watershed, North-West of Iran | 991 | 6152±1266 | This study |

Figures

Fig. 1 Subcatchment names are blurred and mixed with the division lines. I think you could create a different type of labels such as pin flags, for example.

Sorry for this, visibility has improved, and the names have been changed accordingly.

The scale could be smaller, and a black-white style could fit better.

Done, thank you.

Instead of displaying the lake on the magnified Kurdistan province map, you could display the catchment limits.

We added the catchment as suggested by the reviewer (see below).

Streams or rivers. These terms are pretty much interchangeable, but according to your scale, you have a stream with a length of 2000Km.

[Figure]

Figure 1: The location of the Zarivar Lake watershed in Kurdistan Province, Iran.

It will definitely help to visualise your study if you could include some pictures of the forest and vineyards.

Thanks for the suggestion! Pictures were added to the Fig. 2 (see below).

Fig. 2 At least in my figure, the legend does not fit with the symbols. Furthermore, it is quite difficult to discern the separation between points. When plotting in Surfer (I think you used this software) the colours suffer a small colour change. Please try to modify your legend accordingly.

Is it true that your stream/river ends before reaching the outlet?

The height axis was not specified, and the meters cannot be seeing. If it is difficult to modify this in the software used, you could try to modify it by using additional graphical software.

Sorry for these and thanks a lot for these suggestions; we have considered all proposed modifications (see below).

The colour legend about the height is not specified. However, it would be nice to see the land use map over the DEM. It can be done in Surfer.

Since the land-use has already been shown in fig 1, we decided not to add to the second map as well. Hope you agree.

[Figure]

**Figure 2: The location of the reference site and the study hillslopes at Z3 sub-watershed.**

| Land use | Average of slope (%) | Slope length (m) | Slope aspect |
|---|---|---|---|
| Forest | 30 | 200 | NE |
| Vineyard | 24 | 180 | NE |

Fig. 5 I think you should use the same colour scale for both maps. Thus, you can easily compare them.

Thanks! The legend has been modified (see below).

[Figure]

**Figure 5: Total precipitation in April and May 1986 (from CHELSA dataset; Climatologies at high resolution for the earth's land surface areas).**

Fig. 6 What are the green dots? Please show it also in the legend.

They have been added to the legend (see below).

[Figure]

**Figure 6: Ratio of $^{240}$Pu/$^{239}$Pu of samples at reference site and the forested hillslope.**

Fig. 7 It is already nice, but you could improve by giving the equation the colour f the lines. However, do it carefully because maybe the green colour is too light to be correctly visualised.

As suggested by Reviewer #2, the color has been changed in the figure.

Fig. 8 A little bit of colour here it would be nice.

The colors have been altered (see below).

[Figure]

**Figure 8: FRNs derived soil redistribution rates in a forested transect using MODERN and Diffusion and Migration model (DMM) (negative values indicate erosion, whereas positive values indicate deposition).**

Fig. 9 Same here, but just as a piece of advice.

Thanks, the color of FRNs' depth distributions at the vineyard hillslope has been different from those of the forest site (see below).

[Figure]

**Figure 9: Depth distribution of $^{137}$Cs (a), $^{210}$Pb$_{ex}$ (b) and $^{239+240}$Pu (c) in an erosional site in forested hillslope and $^{137}$Cs (e) and $^{210}$Pb$_{ex}$ (f) at an erosional site in vineyard hillslope.**

Specific comments and technical corrections

Apart from the general comments, I have added some specific comments about the text and the figures. The authors might find them useful to improve their manuscript. "In my opinion", the inclusion of these points would significantly increase the audience interest of this manuscript that presents an interesting study. Here I have attached some typing error or minor suggestions to improve the text:

Thanks so much for your time and effort to improve this manuscript. Much appreciated.

Line 38-39 I think this could better fit. "The transition/conversion from natural covers to cultivated lands have increased drastically."

It has been modified in the MS.

Line 40 "more rapid than ever before" I think this is too much to state. Maybe something like this "the las century/or centuries" could do the job.

As suggested, it has been added to MS.

Line 55 "and the subsequent increase in soil erosion."

This has been revised in the MS.

Line 55-56 This statement is too general. In the northern part of the Mediterranean region, especially in mountain agroecosystems, recent land-use changes produced just the inverse trend due to the land abandonment (that it is also a land use change). You could find many recent manuscripts using 137Cs and also 210Pb techniques that describe the issue, especially from Spanish an Italian catchments.

Thanks so much; the sentence has been removed.

Line 63, 65, 69 just as a recommendation, I would prefer if you don't use the words "e.g." that much.

They have been corrected in the MS.

Line 100 Did the authors missed a reference?

Thank you! The reference (i.e. Auerswald, 1995) has been added.

Line 104 runoff?

We corrected this mistake in the MS.

Line 111 "deforestation on and soil". Could it be mistyped?

Thanks! It has been corrected accordingly.

Line 120 Figure two shows nothing related to 12 sub-watersheds. I think you should rewrite this paragraph to make it clearer for the readership. It is not very clear.

The sentence has been rewritten as follows: "The watershed was sub-divided into 12 sub-watersheds named Z1 to Z12 (Fig. 1); one of these watersheds called "Z3" was selected for this study."

Line 121 It would be nice to include the data of max and min altitude here. You have graphically included in the figure, but it has not been specified.

They have been added to the text accordingly (see below).

"The area of the sub-watershed is about 2.97 Km2 with an average altitude of 1518 m a.s.l. (ranging from 1292 to 1876 m a.s.l.) and the landscape topography ranges from gentle to very steep slopes."

Line 140 Besides,?

As suggested by Reviewer, it has been corrected in the MS.

Line 143-15 A fragment could be missing. For what have they been constructed? I think it is obvious, but it would be nice to specify it; thus, making your point clearer to the readership.

The reason has been added as follows: "However, conservation measures like building terraces and check dams have been taken in some parts of the watershed to mitigate sediment entering the lake during the last decade"

Line 150 Keep the same style. Here you've used italics.

It has been corrected in the MS.

Line 247 This is an important part of the work. Thus, further description is needed.

We added a clearer description: "At our study site, the average atomic ratio of 240Pu/239Pu was $0.184 \pm 0.020$ ranging from 0.121 to 0.262 (Fig. 6). In addition, to determine the proportion of the 137Cs Chernobyl input at our study site, the mean activity ratio of 137Cs/239+240Pu from global originated fallout was employed using the value reported by Hodge et al. (1996) at 38.4 (as of 1 July 1994) after values being decay-corrected to 1.10.2016, this factor reduced to 22.9. Following this approach, the global derived 137Cs inventory should correspond to 22.9 multiplied by the 239+240Pu reference inventory. The Chernobyl 137Cs was calculated by subtracting the global inventory contribution from the reference site inventories (which includes both global and Chernobyl fallout)."

Line 285 It would be nice to see a correlation matrix between all the properties.

It does not seem valid since the number of samples is very limited (six and seven points in the vineyard and forested hillslope, respectively).

Line 323 You cite a strong study about it, but you could also include the correlation of the data presented in Table 1. It will be only around 0.5 but still could support your discussion.

Thanks, the correlation has been added. Also, newly added references have been considered (see below and table 1 shown above).

"Using data in Table 1, also was observed a significant correlation between mean reference inventories of [137]Cs and MAPs ($p<0.01$, y =5.25x -500.02; $r^2 \sim 0.72$; y and x are [137]Cs inventory and MAP, respectively), which reinforced this hypothesis."

---

## Author Comment (AC3)

**Answers to Reviewer#3,**

The paper shows evidence of an increase in soil erosion rates following a change in land use. More specifically, the authors documented erosion rates in areas covered by forests and in areas where the same forests were replaced by vineyards. The erosion rates for the vineyards were approximately five times those documented for the forests.

The work is based on measurements made in two adjacent and similar hillslopes located in the Zarivar Lake watershed, Kurdestan Province, Iran. The authors selected these sites because this area, like many others in the country, was affected by this land-use conversion several years ago.

The sampling campaigns consisted of collecting a number of soil samples for Cs-137, excess Pb-210 and Pu-239,240 analyses. The area was affected by the Chernobyl accident. In this respect, the authors calculated the Cs-137 Chernobyl component using the ratio Cs-137/Pu-239,240 assuming that for bomb fallout this ratio is 38.4 (as suggested by Hodge, 1996). The proportion of the Chernobyl component was estimated to be ca. 50% of the total fallout (including the bomb-derived).

In general, the manuscript is acceptable and can be considered for publication. However, I have some comments about the methods and about some of the statements provided by the authors. These are as follows.

**Thank you for your positive feedback and your time and effort to improve this manuscript. Much appreciated.**

Lines 76-77 - The authors indicated that the global-derived 137Cs fallout ranges between 160 and 3200 Bqm-2 depending on latitude (UNSCEAR, 1969; Garcia Agudo, 1998). Here the authors should say that these two boundary values are referred to 1996 otherwise they need to be decay corrected. Please add some comments.

However, there are many exceptions to the map published by Garcia Agudo because 137Cs fallout is very sensitive to precipitation amount as well. See examples in Canada (Mabit et al., 2002), Spain (Navas et al., 2007; 2017), USA (Arnalds et al., 1989), India (Mishra and Sadasivan, 1972), Italy (Porto et al., 2009) etc. where the fallout basic line was correlated with the rainfall amount. Please consider to add some comments and citations about it.

Our aim here was to highlight the expected range of the global-derived 137Cs fallout at reference sites. As suggested, all mentioned literature was investigated. The 137Cs inventory at reference sites and the decay corrected values (if the year of sampling is not mentioned in the publication, we select the publication date) are summarized in the table below. We could not find any unusually high values except for Spain (Navas et al., 2017), which is a site characterized by high Mean Annual Precipitation (MAP) and Italy (Porto et al., 2009). Please also note that "Fallout 137Cs originating from the Chernobyl accident, which affected many areas in Europe, increased the existing bomb-derived inventories by several orders of magnitude in some locations (Mabit et al., 2008)" (also see map below). Thus, the sentence was corrected as follows:

"Generally, the global pattern of global-derived 137Cs fallout indicates that inputs are distributed between 160 and 3200 Bqm-2 (decay corrected to 1996) depending on latitude (UNSCEAR, 1969; Garcia Agudo, 1998)."

| Location
(Reference)                | 137Cs inventory at
reference site (Bq
m–2) | 137Cs inventory at
reference site
decay corrected to
1996 (Bq m–2) | Remarks                                                                         |
|----------------------------------------|--------------------------------------------------|-----------------------------------------------------------------------------|---------------------------------------------------------------------------------|
| Canada (Mabit et al., 2002)            | 2650                                             | ~3043                                                                       |                                                                                 |
| India (Mishra and
Sadasivan, 1972). | e.g. 40 (?)                                      | ?                                                                           | The reported 137Cs
inventories are not for
reference site s |
| USA (Arnalds et al., 1989)             | 1300-7480                                        | ~1100-6360                                                                  | The reported 137Cs
inventories are not for
reference sites  |
| Spain (Navas et al., 2007)             | 1710                                             | ~2203                                                                       | Central Ebro valley, Spain
(MAP=300-500mm)                                   |
| Spain (Navas et al., 2017)             | 4500                                             | ~7300                                                                       | Central Spanish Pyrenees
(MAP=800-2000mm)                                    |
| Italy (Porto et al., 2009)             | 2805-4682                                        | ~ 3786- 6319                                                                | Southern Italy
(MAP=990-1302 mm)                                             |